

# Stable isotopes track the ecological and biogeochemical legacy of mass mangrove forest dieback in the Gulf of Carpentaria, Australia

Yota Harada[1], Rod M. Connolly[1], Brian Fry[2], Damien T. Maher[3,4], James Z. Sippo[3], Luke C. Jeffrey[3], Adam J. Bourke[5], Shing Yip Lee[6]

[1]Australian Rivers Institute – Coast and Estuaries, and School of Environment and Science, Griffith University, Gold Coast, Queensland, Australia
[2]Australian Rivers Institute, Griffith University, Nathan, Queensland, Australia
[3]Southern Cross Geoscience, Southern Cross University, Lismore, New South Wales, Australia
[4]School of Environment, Science and Engineering, Southern Cross University, Lismore, NSW, Australia
[5]College of Engineering, Information Technology and the Environment, Charles Darwin University, Northern Territory, Australia
[6]Simon F.S. Li Marine Science Laboratory, School of Life Sciences, and Earth System Science Programme, The Chinese University of Hong Kong, Shatin, Hong Kong SAR, China

*Correspondence to*: Yota Harada (yota.harada@griffithuni.edu.au)

**Abstract.** A combination of elemental analysis and stable isotope analysis (SIA) was used to assess and monitor C, N and S cycling of a mangrove ecosystem that suffered mass dieback of trees in the Gulf of Carpentaria, Australia in 2015-16, attributed to an extreme drought event. Three field campaigns were conducted over a period from 2016 to 2018, at 8, 20 and 32 months after the event. Samples including invertebrates, mangroves, and sediment were analysed for CNS elemental and isotopic compositions including compound-specific stable isotope analysis (CSIA) of amino acid carbon. Samples collected from the impacted ecosystem were enriched in $^{13}$C, $^{15}$N and $^{34}$S relative to those from an adjacent unimpacted reference ecosystem, likely indicating lower mangrove carbon fixation, lower nitrogen fixation and lower sulfate reduction in the impacted ecosystem. For example, invertebrates representing the feeding types of grazing, leaf feeding, and algae feeding were more $^{13}$C enriched at the impacted site, by 1.7 – 4.1‰ and these differences did not change over the period from 2016 to 2018. The CSIA data indicated widespread $^{13}$C enrichment across five essential amino acids and all groups sampled (except filter feeders) within the impacted site. Mangrove seedling and sapling populations increased substantially from 2016 to 2018 in the impacted forest, suggesting recovery of the mangrove vegetation. Recovery of CNS cycling, however, was not evident even after 32 months, suggesting a biogeochemical legacy of the mortality event. Continued monitoring of the post-dieback forest would help to predict the long-term trajectory of ecosystem recovery. In such long-term monitoring programs, SIA that can track biogeochemical changes over time can help to detect underlying biological mechanisms that drive changes and recovery of the mangrove ecosystem. To gain further insight, our use of CSIA can help show feeding dependencies in mangrove food webs and their response to disturbances.



## 1 Introduction

Stable isotope analysis (SIA) is a powerful tool for environmental assessment and monitoring that provides information about biogeochemical source and processes over time. As elements such as carbon (C), nitrogen (N) and sulfur (S) circulate in the biosphere, stable isotopic compositions of $^{13}C/^{12}C$, $^{15}N/^{14}N$ and $^{34}S/^{32}S$ change in predictable ways due to mixing and fractionation, giving insights into sources and cycling of these elements (Fry 2006). SIA has been widely used in mangrove ecosystem studies to better understand food web interactions (Bouillon et al., 2008; Larsen et al., 2012; Bui and Lee, 2014; Abrantes et al., 2015), mangrove nutrient uptake (McKee et al., 2002), mangrove water use (Santini et al., 2015; Hayes et al., 2019), cycling of C (Maher et al., 2013a; Maher et al., 2017; Sasmito et al., 2020),  N (Fry and Cormier, 2011), S (Raven et al. 2019), and greenhouse gas emissions (Maher et al., 2013b).

Low frequency, high intensity weather events, such as droughts, tropical cyclones, heatwaves and climatic extremes can cause mass mortality of foundation species such as mangroves (Sippo et al., 2018), saltmarshes (Silliman et al., 2005), seagrasses (Thomson et al., 2015), kelps (Wernberg et al., 2016) and corals (Hughes et al., 2017). The frequency and intensity of extreme climatic events are expected to increase due to climate change (Coumou and Rahmstorf, 2012; Stott, 2016). In 2015-16, an extensive area (>7000 ha) of mangrove forest along ~1,000 km of coastline in the Gulf of Carpentaria, Australia, experienced severe dieback, an event associated with the climatic extreme of drought (Duke et al., 2017).  As mangroves show characteristics of pioneer species (Tomlinson, 2016), large-scale disturbances have likely played an important role in their evolution. However, the processes, rates and patterns of recovery from disturbances are still largely unknown. In most cases, recovery of mangroves primarily relies on the recruitment of seedlings (Smith et al., 1994; Krauss and Osland, 2019). Disturbances in mangrove forests not only affect recruitment, but can also change the cycling of C, N and S. Loss of mangrove trees and root structures can change organic matter inputs, sediment oxygenation and degradation of sediment organic matter. These changes alter overall sediment conditions, with consequences for benthic assemblages (Sweetman et al., 2010; Bernardino et al., 2018; Harada et al., 2019), sediment C and N stocks (Adame et al., 2018), microbial assemblages, and associated nutrient processes e.g. nitrogen fixation and sulfate reduction (Sjöling et al., 2005).

Traditional methods to evaluate structure and functioning of ecosystems include measures of species composition, but these field assessments can be time-consuming and expensive and may not provide enough quantitative information about system functioning (e.g. Kling et al.,1992). SIA of ecosystem components is a powerful way of quantitatively evaluating functional aspects of element cycling and the health of ecosystems (e.g. integrity of the food web; Fry, 2006). SIA of total organic matter (''bulk'') have become widespread due to the relative ease and low cost of sample preparation and analysis (Fry, 2006). Compound-specific stable isotope analysis (CSIA) is increasingly employed as a complementary tool to bulk SIA. For instance, while bulk SIA of C, N and S provide an overview of food webs (Bouillon et al., 2008), CSIA of amino acids (AAs) help measure details of organic matter cycling (Ishikawa et al., 2018; Larsen et al., 2013; Ohkouchi et al., 2017)




We investigated changes in C, N and S cycling associated with the Gulf of Carpentaria mangrove forest dieback (Duke et al., 2017), using a combination of traditional ecological survey techniques, bulk SIA, and CSIA of amino acid carbon. We hypothesised that the mortality of mangrove foundation species has changed the overall circulation of C, N and S elements and these biogeochemical changes would most likely be reflected in $\delta^{13}C$, $\delta^{15}N$ and $\delta^{34}S$ values of mangrove ecosystem
components such as mangrove plants, sediment and associated animals. We also tested the hypothesis that these isotopic compositions changed over time with the recovery of mangrove vegetation. $\delta^{13}C$, $\delta^{15}N$ and $\delta^{34}S$ values were measured for samples including mangroves, sediment and invertebrates collected in a comparative setting of impacted mangrove forest site and an adjoining unaffected reference forest site in the Gulf of Carpentaria, Australia.

## 2 Material and Methods

### 2.1 Study site

Over 7,000 ha of mangroves along ~ 1,000 km of the Gulf of Carpentaria coastline in Australia experienced mass mortality during the summer in 2015-16, (Duke et al., 2017), the most extensive mangrove forest dieback ever recorded due to natural causes (Sippo et al., 2018). At the same time, there were coinciding mangrove mass mortality events in Exmouth, Western Australia (Lovelock et al., 2017) and Kakadu National Park, Northern territory (Asbridge et al., 2019). The climate in the
Gulf region is wet-dry tropical with mean annual precipitation ranging from approximately 600 to 900 mm. Dry conditions prevail for six to eight months and most rainfall occurs between December and March (Bureau of Meteorology, see www.bom.gov.au). The climatic conditions limit the extent of mangroves in the region (Asbridge et al., 2016). The dieback was most likely linked to a weak monsoon (low rainfall), combined with high vapor pressure deficit, and El Niño–Southern Oscillation-induced low sea-levels (Duke et al., 2017; Lovelock et al., 2017; Harris et al., 2017; Harris et al., 2018). These
conditions most likely resulted in hypersalinization and caused accumulative hydric, thermal and radiant stresses (Duke et al., 2017; Lovelock et al., 2017; Harris et al., 2017; Harris et al., 2018). The event led to the widespread death of mangrove trees in the region providing a unique opportunity to test tree mortality effects on biogeochemical and ecological functioning of mangroves and capture recovery patterns.

Three field campaigns were conducted in August 2016 (8 months after the event), August 2017 (20 months after the event) and August 2018 (32 months after the event) in the winter dry-seasons in Karumba, Gulf of Carpentaria, Australia (Fig. 1A). Some local factors (e.g. river influence) may have kept some mangroves from dying back in the region. A forest that had suffered dieback (impacted) on the east of Norman river outlet and an adjoining unaffected forest (unimpacted) on the west, provided the setting for comparisons. *Avicennia marina* was the dominant mangrove species. In order to assess differences
between the two forests (impacted vs. unimpacted), as well as to capture trends from across the intertidal zone and to ensure that the physical-oceanographic conditions between the two forests were as similar as possible, three sampling transects (2 to





2.5 km apart) were set for each forest with the length of each transect being approximately 200 m (Fig. 1B, C). Due to logistical constraints and the presence of saltwater crocodiles, fieldwork was restricted to daytime, low tide and dry seasons.

## 2.2 Samples

Three field campaigns were carried out in August 2016, 2017 and 2018. Since our focus was to measure recovery of mangrove vegetation and food-web, we monitored mangrove sapling/seedlings and stable isotopes of invertebrates during the period from 2016 to 2018. Mangrove and sediment samples were also collected but they are limited to 2018 (Table 1). Some of the SIA samples including invertebrates, mangrove and sediment collected in 2017 were used to measure the initial dieback reported in Harada et al. (2019).


During each field campaign, four common mangrove macroinvertebrate groups with different feeding modes were collected from each forest including a leaf-eating crab (*Parasesarma molluccensis* and/or *Episesarma* sp.), an algae-eating (deposit-feeding) crab (*Tubuca signata*), a grazer gastropod (*Telescopium telescopium*) and a filter-feeding bivalve (*Saccostrea sp,* an oyster). For each feeding group, 3 to 5 individuals at each of the sampling transects (n=3) within the forest were collected

and muscle tissues were pooled for SIA.

In 2018, we further divided each transect into five zones (50m apart), namely forest edge (landward), high, mid, low and forest edge (seaward). Fully developed green leaves of *A. marina* were collected from at about 1 to 1.5 m height, from 3 to 5 individual trees (1 to 3 leaves per tree) at each zone, stored in plastic containers, then composited. In the impacted site,

regrowth was occurring in some trees, and leaves were collected from this regrowth. Leaf samples were washed thoroughly, rinsed with distilled water and the main vein was removed. Additionally, wood samples (n=2) were collected from each forest. Two to three bulk SIA measurements were made for each wood sample and measurements were averaged. Wood samples were generally very low in S and only one wood sample had sufficient S for $\delta^{34}$S analysis.

In 2018, surface (<0.5 cm) sediment was collected along the intertidal zones of each transect from the forest edge (landward) to forest edge (seaward) and also in the adjacent mudflat. Additionally, in each forest, 0.5 to 20 cm sediment samples (n=6) were collected using a core sampler, 5 cm in diameter and 20 cm deep. For $\delta^{13}$C measurements, the sediment samples were acidified with 1M HCl to remove the inorganic fraction. Microphytobenthos (MPB) samples (n=6) were separated from surface sediment collected at each forest. The separation was achieved through density gradient centrifugation in Ludox

colloidal silica (Sigma) as described in Bui and Lee (2014). The MPB extraction was followed by microscopic examination to confirm that samples mostly contained green cells (i.e. diatoms and filamentous cyanobacteria). Additionally, surface sediment samples were collected from offshore (approx. 1km) using a grab sampler and from the adjacent saltpan (approx. 200m inland from the forest). Offshore water samples (n=3) were also collected and then filtered through glass fiber filters (diameter 44mm, pore size 0.7μm, Whatman GFF) to obtain particulate organic matter (POM).






To estimate mangrove seedling/sapling densities (ind. m$^{-2}$) from each forest and their changes over the two-year period (2016 to 2018), seedling/saplings were counted with 50 x 50 cm quadrats. In this process, photographs were taken in the field (for 2016, n=124 for the unimpacted forest and n=143 for the impacted forest, for 2017, n=161 and n=175, and for 2018, n=80 and n=17, respectively) and then counts of seedlings and samplings were made in the laboratory. The

seedlings/samplings were mostly *A. marina* but also include some *Aegiceras corniculatum*.

### 2.3 Stable isotope analysis

All invertebrate, leaf, sediment and filter samples were stored separately in sealed plastic containers at -20°C until analysis, then dried at 60°C, powdered, homogenized and put in tin capsules for SIA. $\delta^{13}$C, $\delta^{15}$N and $\delta^{34}$S measurements were carried out on an elemental analyzer (Europa EA-GSL, Sercon) coupled to an isotope ratio mass spectrometer (Hydra 20-22,

Sercon) at Griffith University, Brisbane, Australia. Isotope values are reported relative to Vienna Pee Dee Belemnite (PDB), atmospheric N$_2$ (AIR), and Vienna Canyon Diablo Troilite (VCDT) for C, N and S, respectively. Harada et al. (2019), reported $\delta^{13}$C and $\delta^{15}$N values for some of the samples collected in 2017 from the same study location.

The samples collected in 2017 showed substantial differences in $\delta^{13}$C values between the two forests (Harada et al., 2019)

suggesting that they are representative of the dieback impact. The 2017 samples from each forest including the mangrove leaf (n=2), MPB (n=1), algae feeder (n=3), leaf feeder (n=2 to 3), grazer (n=3), filter feeder (n=2) were further measured for carbon isotopic composition of individual amino acids ($\delta^{13}$C$_{AA}$). For this CSIA, 8 mg (for animal tissues) or 30 mg (for plant tissues) of sample materials were transferred to borosilicate vials with heat and acid-resistant caps. They were then flushed with N$_2$ gas, sealed and hydrolysed in 0.5mL (animal tissues) or 2mL (plant tissues) of 6M HCl at 150°C for 70 minutes,

then dried in a heating block at 60°C under a stream of N$_2$ gas. The dried samples were derivatised by methoxycarbonylation as described by Walsh et al. (2014). Amino acid derivatives were separated by a Trace GC Ultra gas chromatograph (Thermo Scientific) using a DB-23 column, Agilent, 30m x 0.25mm, 0.25μm film at the stable isotope facility at the University of California (Davis, CA, USA). The GC was interfaced with a Delta V Plus isotope ratio mass spectrometer via a GC IsoLink (Thermo Scientific). L-Norleucine was used as an internal standard and to calculate provisional values.


Pure AAs mixtures with calibrated $\delta^{13}$C were co-measured. One mixture was used for final calibration and others were for the scale normalisation standard and the primary quality assurance standard (unused in corrections). Two working standards were co-measured as secondary quality assurance materials. Exogenous carbon was accounted by the method detailed by Docherty et al. (2001). Following these processes, $\delta^{13}$C values were determined for 10 AAs (Gly, glycine; Asx, aspartic

acid/asparagine; Pro, proline; Glx, glutamic acid/glutamine; Ala, alanine; Lys, lysine; Ile, isoleucine; Leu, leucine; Phe, phenylalanine; and Val, valine). Met, methionine; His, histidine; and Hyp, hydroxyproline were at or below the limit of



quantitation (LOQ) for some samples. Since we were interested in $\delta^{13}C$ values of essential amino acids ($\delta^{13}C_{EAA}$), we only report $\delta^{13}C$ values of Lys, Ile, Leu. Phe and Val.

## 2.4 Data analysis

All statistical analyses were undertaken in R version 3.4.3 with RStudio interface version 1.1.414. Differences among group means were explored with ANOVA, but for the count data i.e. seedling/ sampling populations, generalized linear model (GLM) with Poisson distribution was used. Before performing ANOVA, the assumptions of homogeneity of variance and normality were tested using Levene's and Shapiro-Wilk's tests, respectively. Two way ANOVA was used to test the effects of time (year) and forest type (unimpacted and impacted) on stable isotope values of invertebrates. To explore $\delta^{13}C$ patterns

among five EAAs, $\delta^{13}C_{EAA}$ values were normalised to the respective sample means following the procedure of Larsen et al. (2009) as follows:

$$Norm(\delta_{EAA}) = \delta_{EAA} - \mu \quad\quad\quad (1)$$

where $\mu$ represents the mean value of all five EAAs (Ile, Leu, Lys, Phe and Val) in the sample. PERMANOVA was performed to test if the pattern of $\delta^{13}C$ among five EAAs of samples differ between the forests. In this analysis, the

normalized $\delta^{13}C_{EAA}$ dataset was used and the Euclidean distance was used as distance metric. Permutation test of multivariate homogeneity of dispersions was performed to check whether dispersions around the centroids are similar between the two forests. All statistical tests used a significance criterion of $\alpha$=0.05.

## 3 Results

### 3.1 Mangroves

In 2016, approx. 8 months after the mangrove mortality event, at the impacted site, mangrove seedling and sapling populations were lower than the unimpacted site (GLM df = 1, estimate 3.75, p < 0.001) but significantly increased throughout the period from 2016 to 2018 (GLM df = 2, estimate 0.01, p < 0.001; Fig. 2). C, N and S elemental compositions (%) of the dominant mangrove species, *A. marina* did not differ greatly between the two sites, but the isotopic compositions varied considerably in 2018 (32 months after the dieback; Table 2). The $\delta^{13}C$ values (mean ± SD) of green leaves harvested

from *A. marina* trees, were significantly higher (i.e. more $^{13}C$-enriched) in the impacted site (-25.8 ± 1.0‰) than the unimpacted site (-28.4 ± 1.5‰; ANOVA $F_{1, 28}$ = 32.9, p < 0.001; Fig. 3A, Table 2). The $\delta^{15}N$ values varied more in the impacted site (ranged from -0.9 to 6.7‰) than in the unimpacted site (ranged from 2.9 to 6.2‰; Fig. 3A). The $\delta^{34}S$ values were generally higher in the impacted site (13.5 ± 5.4‰, range 7.7 to 23.3‰) than the unimpacted site (12.6 ± 5.6‰, range 5.0 to 21.9‰). Leaf $\delta^{34}S$ values became more $^{34}S$ depleted from higher to lower intertidal zones in the impacted site

(ANOVA $F_{4, 10}$ = 5.56, p = 0.013; Fig. 3B). This pattern was weaker in the unimpacted site, but leaf $\delta^{34}S$ values substantially varied across intertidal zones (ANOVA $F_{4, 10}$ = 6.48, p = 0.007; Fig. 3B). Leaf $\delta^{13}C$ and $\delta^{15}N$ values did not display such





patterns along the intertidal zones. Leaf C, N and S (%) did not show any clear trends among two forests and along transects. Yellow leaves generally had a higher S content (~1.2%) than green leaves (0.5 to 0.7%) (Table 2). Wood samples generally had very low N and S contents, significantly lower than the leaves. However, wood samples from the impacted site had a relatively high S content (0.31 ± 0.37%) and had a $\delta^{34}S$ value of 16.6‰ (Table 2).

## 3.2 Sediment

For the surface (0 - 0.5 cm) sediment collected in 2018 (32 months after the dieback), TOC (%) differed significantly between the two forests with the values (mean ± SD) of 2.02 ± 1.16% for the unimpacted site and 1.06 ± 0.37% for the impacted site (ANOVA $F_{1, 28}$ = 12.75, P = 0.001), suggesting that the surface sediment from the impacted site contains ~48% lower TOC relative to those of the unimpacted site (Table 3). The pattern was consistent across the intertidal zones (Fig 4A). The surface sediment TN (%) was also significantly lower for the impacted site (0.09 ± 0.03) than the unimpacted site (0.15 ± 0.06%; ANOVA $F_{1, 28}$ = 9.32, P = 0.005). TOC of mudflat (<0.5 cm) sediment collected adjacent to the two forests also differed significantly with those of impacted site being lower (0.58 ± 0.18%) than the unimpacted site (1.02 ± 0.08%; ANOVA $F_{1, 4}$ = 14.54, P = 0.019). TN (%) of mudflat (<0.5 cm) sediment was also significantly lower for the impacted site (ANOVA $F_{1, 4}$ = 9.81, P = 0.035). TOC (%) of 0.5 – 20 cm sediment did not differ significantly between the two sites with the values of 1.83 ± 0.73% for the unimpacted and 1.29 ± 0.55 % for the impacted site (ANOVA $F_{1, 10}$ = 2.07, P = 0.181). TN (%) of 0.5 – 20 cm sediment also did not differ significantly between the two sites with the values of 0.09 ± 0.03% for the unimpacted and 0.12 ± 0.04% for the impacted site (ANOVA $F_{1, 10}$ = 3.02, P = 0.113).

The $\delta^{13}C$ values of surface (< 0.5 cm) sediment differed significantly between the two sites with those from the impacted site showing higher values (-21.8 ± 1.0‰) than the unimpacted site (-24.3 ± 1.2‰; ANOVA $F_{1, 28}$ = 22.48, P < 0.001). This pattern was consistent across the intertidal zones with $\delta^{13}C$ values from the impacted site becoming similar to those from the adjacent mudflat (Fig. 4b). However, those of 0.5 to 20 cm sediment did not differ significantly with the values of -24.4 ± 0.5‰ for the impacted site and -25.2 ± 0.9‰ for the unimpacted site (ANOVA $F_{1, 10}$ = 3.92, P = 0.076). Surface (< 0.5 cm) sediment collected in the adjacent mudflat did not display a significant difference in $\delta^{13}C$ values between the two sites with the values of -21.2 ± 0.9‰ for those collected adjacent to the unimpacted forest and -21.8 ± 0.8‰ for those collected adjacent to the impacted site (ANOVA $F_{1, 4}$ = 0.64, P = 0.47). The $\delta^{13}C$ value of the surface sediment (-21.8 ± 1.0‰) in the impacted forest was similar to those collected in the mudflat (-21.2 ± 0.9‰) which were also similar to those collected from offshore (-21.8 ± 1.1‰). Those $\delta^{13}C$ values also matched with the $\delta^{13}C$ value of POM collected offshore (-21.5 ± 1.5‰). Surface (< 0.5 cm) sediment collected from adjacent unvegetated saltpan areas also showed similar values (Table 3). MPB extracted from the surface <0.5 cm sediment showed significantly different $\delta^{13}C$ values between the impacted site (-21.5 ± 1.3‰) and unimpacted site (-25.2 ± 1.0‰; ANOVA $F_{1, 10}$ = 28.53, P < 0.001).





### 3.3 Fauna

CNS isotopic compositions of mangrove macroinvertebrates representing algivores, grazers and leaf feeders from the
impacted site were consistently more enriched in $^{13}$C, $^{15}$N and $^{34}$S than their counterparts from the unimpacted forest
throughout the period between 2016 and 2018 (Fig. 5). However, the filter feeding oyster that relies on water column organic
matter showed relatively less differences between the two forests (Fig. 5). Overall, $\delta^{13}$C values of the four feeding groups
range from -23 to -15‰ for the unimpacted site and -20 to -14‰ for the impacted site. The $\delta^{15}$N values ranged from 5.5 to
9.1‰ for the unimpacted site with the fauna in the impacted site having a slightly higher range of 5.6 to 9.5‰. The $\delta^{34}$S
values ranged from 8.2 to 16‰ for the unimpacted site and 13.4 to 21.7‰ for the impacted site. The effect of forest type was
significant for $\delta^{13}$C, $\delta^{15}$N and $\delta^{34}$S values of the algae feeder and the grazer (ANOVA $p < 0.05$). The effect of forest type was
also significant for $\delta^{13}$C and $\delta^{34}$S values of the leaf feeder (ANOVA $p < 0.05$), but was not significant for the $\delta^{15}$N values
(ANOVA $F_{1, 13} = 1.72$, $p = 0.212$). The effect of forest type was not significant for the filter feeder $\delta^{13}$C values (ANOVA $F_{1, 8}$
$= 1.719$, $p = 0.212$), but was significant for the $\delta^{15}$N and $\delta^{34}$S values (ANOVA $p < 0.05$). The effect of time was significant
for the leaf feeder $\delta^{34}$S values, the grazer $\delta^{15}$N values, the filter feeder $\delta^{15}$N and $\delta^{13}$C values (ANOVA $p < 0.05$). Overall, the
$\delta^{13}$C, $\delta^{15}$N and $\delta^{34}$S values of mangrove invertebrates consistently differed between the sites during the period from 2016 to
2018. In 2018, leaf feeder $\delta^{13}$C values, grazer $\delta^{34}$S values and algae feeder $\delta^{34}$S values significantly differed between the
impacted and unimpacted sites (Turkey post hoc test, $P < 0.05$), showing no recovery of the invertebrate fauna $\delta^{13}$C and $\delta^{34}$S
status after 32 months from the dieback event. However, $\delta^{15}$N values started showing matches between the two forests in
2018 (Fig. 5).

### 3.4 Compound-specific isotope analysis of amino acid carbon

The samples collected in 2017 were further measured for carbon isotopic compositions in individual essential amino acids.
$\delta^{13}C_{EAA}$ values corresponded to the bulk $\delta^{13}$C values with the samples from the impacted site consistently showing higher
values than those from the unimpacted forest (Table 4 and Fig 6). The pattern of $\delta^{13}$C among five EAAs (Lys, Ile, Val, Leu
and Phe) for all the consumers did not differ between the two forest types ($p > 0.05$; Table 4 and Fig S1), including the
algae-feeder, leaf feeder, grazer and filter feeder. The $\delta^{13}C_{EAA}$ pattern of mangrove leaves did not differ between the forests,
regardless of the substantial bulk isotope difference between the unimpacted (-26.7 $\pm$ 2.2‰) and impacted (-25.4 $\pm$ 0.1‰)
(Table 4). This isotope pattern was similar for all four feeding groups as well as for the MPB samples (Fig. 6 and Fig S1).





## 4. Discussion

**4.1 Mangroves**

The recovery of mangrove forests from tree mortality events caused by disturbances such as cyclones, generally relies upon the recruitment of seedlings (Smith et al., 1994; Krauss and Osland, 2019). Subsequently, degraded habitats with a reduced seed pool, production and delivery, e.g. by habitat fragmentation, may show slower forest recovery (Milbrandt et al., 2006). The establishment of seedlings may also be inhibited by persistent inundation due to a decreased sediment elevation (Cahoon

et al., 2003; Asbridge et al., 2018). Although mangroves are resilient ecosystems and may recover quickly from natural disturbances (Sherman et al., 2001), in some cases recovery may take more than 10 years (Imbert et al., 2000). Further, mangrove degradation may be followed by fast colonisation of non-mangrove herbaceous species (McKee et al., 2007; Rashid et al., 2009), e.g. succulent saltmarsh (Mbense et al., 2016). In the impacted Gulf of Carpentaria mangrove forest, the density of mangrove seedling/samplings significantly increased throughout the period from 2016 to 2018, suggesting that

recovery is starting to occur in some areas within 32 months after the dieback (Fig. 2).

The substantial differences in CNS isotopic compositions in *A. marina* occurring between the two sites, suggested differences in the environmental conditions and biogeochemical processes that were possibly associated with the mangrove mortality effect. The leaf $\delta^{13}C$ values in the impacted forest were relatively enriched in $^{13}C$. This C isotope pattern may be

due to reduced stomatal conductance that causes lower internal carbon dioxide concentrations and lower carbon isotope fractionation (Farquhar et al., 1989; Lin and Sternberg, 1992a; Lin and Sternberg, 1992b). $^{13}C$-enriched leaf $\delta^{13}C$ values can also be associated with increased carboxylation efficiency associated with higher nutrients, .e.g. N in leaves (Cordell et al., 1999) and thicker leaves with higher internal resistance to carbon dioxide diffusion. Younger leaves can show higher $\delta^{13}C$ values than aged leaves due to $^{13}C$ enriched fractions (e.g. carbohydrates) transported from older autotrophic leaves to more

heterotrophic young leaves (Werth et al., 2015). Leaves exposed to full sun can show higher $\delta^{13}C$ values than shaded leaves (Farquhar et al., 1989). $\delta^{13}C$ values can vary among different plant tissues in *A. marina* (Kelleway et al., 2018). Leaf N (%) did not differ between the two forests, suggesting that the two sites may have similar plant N availability. Previous studies show additions of nutrients such as N and/or P did not play a considerable role in mangrove leaf $\delta^{13}C$ variations (McKee et al., 2002), but salinity played an important role (Lin and Sternberg, 1992b). For example, leaf $\delta^{13}C$ values of *A. marina* at a

lower salinity site was relatively depleted in $^{13}C$ and averaged about -31‰ (Kelleway et al., 2018). Overall, $^{13}C$-enriched leaf $\delta^{13}C$ values in the impacted forest likely suggest that there are chronic stresses associated with the dieback event that reduced stomatal conductance. Such environmental stresses may include hypersalinization of sediments and hydric, thermal and radiant stresses following mangrove losses (e.g. canopy loss).

Leaf $\delta^{15}N$ values varied more in the impacted (ranged from -0.9 to 6.7‰) than in the unimpacted forest (ranged from 2.9 to 6.2‰) (Fig. 3), but the means were similar (4.3 ± 1.9‰ for the impacted and 4.4 ± 0.8‰ for the unimpacted), suggesting



that two sites have similar background $\delta^{15}N$ conditions. Generally, leaf $\delta^{15}N$ varies due to N sources, microbial processes that enrich or deplete $^{15}N$ in soil or water, and isotope fractionation during plant N uptake. Previous studies showed that in pristine mangrove forests, leaf $\delta^{15}N$ values generally range around -2‰ to 3‰ (Fry and Smith, 2002; Smallwood et al.,

2003). Such low $\delta^{15}N$ values may reflect long-term N fixation inputs (e.g. around 0‰) (Fogel et al., 2008) and marine nitrate inputs (Dore et al., 2002). Much higher $\delta^{15}N$ values (>10‰) may be associated with anthropogenic N inputs (Fry and Cormier, 2011). Our sites showed moderate $\delta^{15}N$ values (about 4‰), suggesting that in addition to N fixation inputs and marine N inputs, there may be considerable microbial $^{15}N$ enrichment in dissolved inorganic nitrogen pools of ammonium and nitrate. The higher variability in leaf $\delta^{15}N$ in the impacted forest suggests higher variability in processes affecting the

$\delta^{15}N$ status of available N. For example, changes to sediment conditions following the dieback affected microbial processes of N, whereas the unimpacted forest has more stable N pool, processes and inputs. Isotope fractionation during plant N uptake may also be an explanation for leaf $\delta^{15}N$ variability (Fry et al., 2000), but such fractionation is poorly known for mangroves. A study reported that additions of P nutrients increased N demand and decreased $^{15}N$ fractionation (McKee et al., 2002), however as we did not measure P, we could not determine whether this was the case.


Leaf $\delta^{34}S$ values differed considerably between the two forests, with the impacted forest generally having higher values (13.5 ± 5.4‰, range 7.7 to 23.3‰) than the unimpacted forest (12.6 ± 5.6‰, range 5.0 to 21.9‰). Leaf $\delta^{34}S$ values showed trends along the six transects, with values decreasing from the upper to lower intertidal zones (Fig. 3B). Based on previous studies, mangrove leaf $\delta^{34}S$ values generally vary between -20 to 20‰ (Okada and Sasaki, 1995, 1998; Fry and Smith, 2002). Higher

$\delta^{34}S$ values are likely associated with seawater sulfate, which is $^{34}S$ enriched (i.e. 21‰) and due to a large isotope fractionation (up to 70‰) during sulfate reduction (Kaplan and Rittenberg, 1964). Lower $\delta^{34}S$ values are likely associated with sedimentary sulfide-S that is $^{34}S$ depleted, for example, -21‰ (Okada and Sasaki, 1995). Leaf $\delta^{34}S$ values of around 14 to 18‰ suggest mangrove incorporations of seawater sulfate-S ($\delta^{34}S$, ~ 21‰), with only a small isotopic fractionation occurring through absorption and assimilation steps (Okada and Sasaki, 1995). Plants generally show $\delta^{34}S$ values slightly

lower than source sulfate-S by an average of -1.5‰ (Trust and Fry, 1992). Low leaf $\delta^{34}S$ values, for instance, the lowest value of 5‰ found in the unimpacted site - suggest that the most probable source of this $^{34}S$-depleted S is sulfide oxidation, followed by mixing with seawater sulfate.  Low $\delta^{34}S$ values in mangrove root vascular tissues may indicate assimilation/oxidation of sulfide, potentially to reduce their toxic sulfide exposure (Fry et al., 1982; Raven et al., 2019), with reported isotope effect of -5.2‰ for non-biological oxidation of sulfide (Fry et al., 1988) and a smaller +1-3 ‰ effect for

anaerobic oxidation of sulfide by photosynthetic bacteria (Fry et al., 1984).

An explanation for our observed $\delta^{34}S$ pattern may be lower plant incorporation of sulfide-S in the impacted site and also in the higher intertidal zones where we expect that mangrove sediment is relatively more oxidised, and the production of



sulfide may be lower due to lower sulfate reduction. High wood δ34S values (16.6‰) and S content (0.31%) in the impacted
forest may suggest degradation of wood by fungi and/or bacteria that incorporate seawater sulfate-S and increase overall
wood δ34S values and S content. Such δ34S patterns have been reported in mangroves (Fry and Smith, 2002)  and saltmarsh
(Currin et al., 1995), where δ34S values of fresh organic matter evolved during degradation steps and gradually increased
towards the δ34S value of seawater sulfate-S (i.e. 21‰).

**4.2 Sediment**

In healthy mangrove forests, the fate of C fixed by primary producers includes burial within the sediment, atmospheric
emissions and outwelling to the ocean (Maher et al., 2018), but how mangrove mortality affects such processes is poorly
understood. In most cases, C within in mangrove sediment decreases following forest loss due to degradation with increased
$CO_2$ emissions (Otero et al., 2017; Adame et al., 2018).  Lower TOC (%) and higher sediment δ13C values in the impacted
forest (Table 3 and Fig. 4) are probably related to sediment C loss and lower mangrove C inputs (i.e. leaf litter) following the
mangrove mortality event. Consistent with this, the sediment N (%) and δ15N data showed a similar pattern. The surface
sediment (0 - 0.5 cm) differed relatively more than the deeper (0.5 to 20 cm) fraction. One explanation for this is that the
surface sediment fraction is generally more aerobic, and therefore remineralization of organic matter occurs more rapidly
(Burdidge, 2011). Sediment δ13C and 15N values can increase during degradation of sediment organic matter (Adame and
Fry, 2016). This isotope pattern has been reported following mangrove loss (Adame et al., 2018). Changes in sediment C and
N may also be associated with root turnover. The MPB δ13C values significantly differed, with those from the impacted
being higher (-21.5 ± 1.3‰) than the unimpacted (-25.2 ± 1.0‰). The higher values probably indicate lower respiratory
inputs of $CO_2$ from mangroves (Maher et al., 2013b). Our findings here are consistent with the finding of Sippo et al. (2019)
that changes to oceanic carbon outwelling rates following mangrove loss are likely associated with a gradual loss of
sediment carbon; similar to our finding of increased sediment δ13C values in the impacted site, an isotope effect may have
been due to loss of sediment mangrove C or replacement of mangrove peats with marine sediment.

**4.3 Fauna**

CNS isotopic compositions of consumers including an algae feeder, a grazer and a leaf feeder from the impacted site were
consistently more enriched in 13C, 15N and 34S. These differences did not change throughout the three sampling of 2016,
2017 and 2018 (Fig. 5). Consistent with the findings from mangrove leaves, MPB and soil, these data suggested substantial
changes in cycling of CNS associated with the mangrove mortality event. Overall, the consumer δ13C values ranged from -
22.9 to -15.2‰ for the unimpacted site, but at the impacted site, consumers were more 13C-enriched (range of -20.0 to -
14.0‰), likely due to the loss of 13C-depleted mangrove organic matter. Consumer δ13C values can change due to changes to
available organic matter, altered feeding dependencies as well as changes to organic matter δ13C values. For example, MPB
δ13C values can change in response to organic matter respiratory inputs. The consumer δ13C values and their ranges at our





study site are fairly consistent with the reported mangrove consumer $\delta^{13}C$ values elsewhere (e.g. Lee, 2000; Bouillon et al., 2002; Demopoulos et al., 2007). Lower consumer $\delta^{13}C$ values near -27‰ are generally associated with mangrove detritus, but in many cases, typical mangrove leaf-eating crab species (i.e. Sesarmidae) can be enriched by about +5‰ from the mangrove detritus (Bui and Lee, 2014). Higher $\delta^{13}C$ values of consumers are generally tied to MPB with a typical isotope effect during assimilation, e.g. $< \sim 1‰$ estimated for small invertebrates (Vander Zanden and Rasmussen, 2001; McCutchan

et al., 2003). Our MPB endmember $\delta^{13}C$ values of -25.2‰ for the unimpacted site and -21.5‰ for the impacted site did not match with the consumer $\delta^{13}C$ values (around -15 to -14‰), suggesting our characterization of MPB endmember $\delta^{13}C$ values was incomplete. This is probably because MPB can vary substantially within mangrove ecosystems (Bouillon et al., 2008) and consumers may be preferentially assimilating more $^{13}C$ enriched fractions of MPB, for example, diatom and/or filamentous cyanobacteria that can range about -15 to -20‰ (Craig, 1953; Fry and Wainright, 1991). The leaf feeder in this

study showed $\delta^{13}C$ values of about -21 to -18‰ and was substantially enriched compared to mangrove leaves (-27 to -25‰), consistent with the findings of Bui and Lee (2014). This also indicated some additional contributions from other sources such as MPB.

Due to difficulties obtaining representative endmembers, mixing analysis using sampled organic matter was not achieved in

this study to quantify feeding dependencies. Alternatively, the consumer data was used to help infer endmembers and assess feeding dependencies, e.g. Riekenberg et al. (2016). POM (-21.1‰) matched with the filter feeders and seemed to be an important food source for the all consumers in both forests. Mangrove leaves (-27 to -25‰) did not seem to be an important source for the consumers with the lowest consumer being -22.9‰ at the unimpacted site and -20.0‰ at the impacted site. The consumers were generally higher than the POM with the highest consumer being -15.2‰ at the unimpacted site and -

14.0‰ at the impacted site, suggesting that there was a substantial contribution from more $^{13}C$ enriched MPB.

Consumer $\delta^{34}S$ values were generally higher in the impacted site (range 13.4 to 21.7‰) than in the unimpacted site (range 8.2 to 16‰) suggesting lower sulfate reduction with decreased sulfide inputs at the impacted site. Fixation of sulfate by phytoplankton occurs with a small isotope effect, around 1 to 2‰ (Fry, 2006), so that $\delta^{34}S$ values of phytoplankton from the

coastal ocean should be close to the seawater sulfate-S value of 21‰. MPB generally have lower $\delta^{34}S$ values than phytoplankton, with reported average values near 10‰ for MPB in a mangrove ecosystem (Harada et al., unpublished), likely due to some use of sedimentary sulfide-S (depleted in $^{34}S$). Our mangrove leaf $\delta^{34}S$ values averaged 13.5‰ for the impacted site and 12.6‰ for the unimpacted site, lower than the seawater sulfate-S. For these reasons, the unimpacted site that had lower consumer $\delta^{34}S$ values could be associated with sulfide inputs with some use of mangrove organic matter and

MPB, whereas the impacted site that had higher consumer $\delta^{34}S$ values are associated more with seawater sulfate. This indicates a change to the S cycling and use of S by plants as well as microbial intermediates in the food web.





The consumer $\delta^{15}$N also indicates possible changes to N cycling, with the consumer in the impacted site generally having higher values than those from in the unimpacted site. The higher $\delta^{15}$N may be associated with degradation of organic matter,
microbial $^{15}$N enrichment in dissolved inorganic N such as ammonium and nitrate during degradation, and $^{15}$N enrichment by microbial intermediates in the food web. The high $^{15}$N may also indicate lower N fixation inputs that typically show low $\delta^{15}$N values, round 0‰. While the $\delta^{13}$C and $\delta^{34}$S values consistently differed between the two forests during the two-year survey, the $\delta^{15}$N values started showing matches between the two forests in 2018. This may suggest recovery of $\delta^{15}$N status to the background conditions, and/or that the recovery of N may be faster than C and S elements. This may be the case as
mangrove ecosystems are generally N limited (Reef et al., 2010), and circulation of N elements is faster than those of C and S elements.

### 4.4 Compound-specific isotope analysis of amino acids

It is considered that environmental resources such as vascular plants and microalgae have a different $\delta^{13}$C pattern ('fingerprint') in AAs due to differing biosynthesis of AAs (Larsen et al., 2009; Larsen et al., 2013). It is also reported that
$\delta^{13}$C patterns are largely unaffected by environmental conditions. For example, $\delta^{13}C_{AA}$ patterns of the marine diatom *Thalassiosira weissflogii* did not respond to changing environmental conditions such as light, salinity, temperature and pH, despite substantial changes in bulk $\delta^{13}$C values (Larsen et al., 2015). A similar isotope pattern was reported for seagrass *Posidonia oceanica* and the giant kelp *Macrocystis pyrifera,* which showed consistent $\delta^{13}C_{AA}$ patterns despite varying season and growth conditions (Larsen et al., 2013). The $\delta^{13}C_{AA}$ patterns in producers, especially those of essential amino acids
($\delta^{13}C_{EAA}$), can reflect consumer tissues with little isotope effect. This occurs because animals obtain EAAs from their diet and EAA fractions are thought to be directly assimilated (McMahon et al., 2010). These general expectations were reasonably met in our $\delta^{13}C_{EAA}$ dataset - that was normalized to means of five EAAs as per Larsen et al. (2009). Normalized $\delta^{13}C_{EAA}$ patterns of our producer samples including mangrove leaves (yellow leaves of *A. marina*) and MPB did not differ between the two sites despite differing environmental conditions and substantial differences in bulk $\delta^{13}$C values (Table 4, Fig
6 and Fig S1). Furthermore, the consumer $\delta^{13}C_{EAA}$ patterns also did not differ between the two sites. (Fig. 6 and Fig S1). These findings did not support changes to feeding dependency following mangrove loss but suggested that the overall differences in the consumer bulk $\delta^{13}$C values were most likely driven by differences in the resource organic matter $\delta^{13}$C values e.g. changes to MPB $\delta^{13}$C values that were likely associated with lower mangrove C fixation/respiratory inputs following the mangrove mortality. Furthermore, such findings indicate that the reported substantial change to the mangrove
benthic faunal assemblage following the mangrove loss (Harada et al., 2019) was probably driven more by modification of physical habitat structure than changes in the use of food resources.



## 5 Conclusions

Reporting rare and extreme biological events can be complicated because in many cases they may occur suddenly, therefore drawing comparisons between pre and post event conditions remains a challenge. Our field investigations using traditional
ecological techniques combined with SIA - measured the initial dieback and also early recovery of an impacted mangrove ecosystem and compared an adjacent unimpacted reference system. Mangrove seedling and sapling populations that increased during the period from 2016 to 2018 (8 to 32 months after the mortality event) in the impacted site, suggest recovery of the mangrove vegetation. This also suggests that the environmental conditions at the impacted site are still conducive for re-establishment of mangroves, allowing recruitment of seedlings and development of regrowth. However,
mangrove leaves collected in the impacted site in 2018 showed relatively higher $\delta^{13}C$ values (-25.8 ± 1.0‰) that are probably associated with continued water stress. Invertebrates from the impacted site representing the feeding types of grazing, leaf feeding, and algae feeding were more enriched in $^{13}C$, $^{15}N$ and $^{34}S$ relative to those from the unimpacted site. For example, they were more $^{13}C$ enriched at the impacted site, by 1.7 – 4.1‰ and the difference did not change over the study period. Overall, our stable CNS isotope data supported the hypothesis that changes to biogeochemical processes occur
following the mangrove mortality. These changes include lower mangrove C fixation/respiration, lower N fixation and lower sulfate reduction. However, our isotope data did not support the second hypothesis that the isotopic compositions change over time with recovery of mangrove vegetation. Recovery of biogeochemical processes was not evident even after two years, suggesting an ongoing impact of the mortality event. An exception was that N cycle recovery may be occurring faster.

Considering that the environmental conditions at the site play an important role in facilitating recolonisation of mangroves, we conceptualise the recovery of the mangrove forest under four different scenarios to give insight into the ecological and biogeochemical consequences of changing forest conditions (Fig. 7): 1) The forest recovers with mangroves being able to recolonise at the site without future perturbations; 2) the forest recovers with future perturbations such as climatic events, for example, mangrove recolonisation is driven by events as such ENSO cycles; 3) the forest does not recover and is
transformed into intertidal mudflats; and 4) the forest recovers partially at the site and in a reduced size and/or is recolonised by other plants such as saltmarshes, e.g., mangroves only recolonise in the lower intertidal zone. Each of these scenarios will have a distinct isotopic trajectory for C, N and S. Continued monitoring of the post-dieback forest would be required to predict the long-term trajectory of ecosystem recovery and how on-going climate change and extreme climatic events affect the recovery of mangroves in the impacted region. In such a long-term investigation, SIA is a powerful tool, capable of
tracking changes in biogeochemical processes over time. As such, it is of great assistance in ecosystem analyses and detecting the underlying biological mechanisms that drive changes and recovery.



*Author contributions.* The study was conceptualized by all authors. Writing was led by YH and contributed to by all. Field surveys were executed by DTM, JZS, LCJ, AJB and YH. Data compilation and analysis was coordinated by YH and

contributed to by all.

*Competing interests*. The authors declare no competing interests.

*Acknowledgments.* We acknowledge support from: YH – Holsworth Wildlife Research Endowment – Equity Trustees

Charitable Foundation & the Ecological Society of Australia; DTM - Australian Research Council (DE1500100581, DP180101285); RMC – Global Wetlands Project. We thank R. Bak and V. Fry (Griffith University) for stable isotope analysis. Field work was assisted by G. Reithmaier, K. Maguire, D. Brown, S. Conrad, A. McMahon, C. Holloway, M. Call, A. Gauthey, G. Balland and J. Kalla. M. Hayes commented on and improved early versions of the manuscript.





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



**Table 1.** Analyses conducted for samples from each field campaign (x = analysed).

| Analysis | 2016 (8 months after dieback) | 2017 (20 months after dieback) | 2018 (32 months after dieback) |
|---|---|---|---|
| Mangrove seedling count | x | x | x |
| Bulk SIA invertebrates | x | x | x |
| Bulk SIA mangrove | | | x |
| Bulk SIA sediment | | | x |
| CSIA invertebrates | | x | |



**Table 2.** Elemental and isotopic compositions of the mangrove *A. marina* (mean, SD) in 2018, 32 months after the dieback.

| Tissue type | Forest | %C | %N | %S | $\delta^{13}C$, ‰ | $\delta^{15}N$, ‰ | $\delta^{34}S$, ‰ | n |
|---|---|---|---|---|---|---|---|---|
| Green leaf | Unimpacted | 39.3, 1.7 | 1.75, 0.37 | 0.54, 0.23 | -28.4, 1.5 | 4.4, 0.8 | 12.6, 5.6 | 15 |
| | Impacted | 40.0, 1.6 | 1.82, 0.45 | 0.73, 0.14 | -25.8, 1.0 | 4.3, 1.9 | 13.5, 5.4 | 15 |
| Yellow leaf | Unimpacted | 41.6, 2.2 | 0.67, 0.23 | 1.18, 0.47 | -26.4, 1.3 | 6.5, 1.1 | 14.6, 7.7 | 3 |
| | Impacted | 41.9, 1.0 | 0.52, 0.05 | 1.18, 0.38 | -26.2, 0.6 | 7.4, 0.2 | 12.5, 2.8 | 3 |
| Wood | Unimpacted | 42.6, 3.0 | 0.38, 0.27 | 0.04, 0.01 | -24.8, 1.8 | 6.1, 2.0 | - | 2 |
| | Impacted | 39.9, 5.7 | 0.40, 0.16 | 0.31, 0.37 | -24.9, 1.0 | 4.7, 1.0 | 16.6 | 2 |




**Table 3.** Elemental and isotopic compositions of sediment in 2018, 32 months after the dieback (mean, SD).

|  | Forest/site | %TOC | %TN | $\delta^{13}$C, ‰ | $\delta^{15}$N, ‰ | n |
|---|---|---|---|---|---|---|
| Mangrove forest, 0 to 0.5 cm | Unimpacted | 2.02, 1.16 | 0.15, 0.06 | -24.3, 1.2 | 2.0, 0.5* | 15 |
|  | Impacted | 1.06, 0.37 | 0.09, 0.03 | -21.8, 1.0 | 2.8, 0.6* | 15 |
| Mangrove forest, 0.5 to 20 cm | Unimpacted | 1.83, 0.73 | 0.12, 0.04 | -25.2, 0.9 | 1.7, 0.5* | 6 |
|  | Impacted | 1.29, 0.55 | 0.09, 0.03 | -24.4, 0.5 | 1.7, 0.3* | 6 |
| Mudflat, 0 to 0.5 cm | Unimpacted | 1.02, 0.08 | 0.11, 0.01 | -21.8, 0.8 | - | 3 |
|  | Impacted | 0.58, 0.18 | 0.07, 0.02 | -21.2, 0.9 | - | 3 |
| Saltpan, 0 to 0.5 cm | Unimpacted | 1.87, 2.03 | 0.19, 0.24 | -18.9, 1.7 | - | 4 |
|  | Impacted | 0.83, 0.07 | 0.07, 0.01 | -20.8, 0.7 | - | 4 |
| Offshore, 0 to 0.5 cm | 1km offshore | 0.70, 0.27 | 0.08, 0.03 | -21.8, 1.1 | - | 5 |
| POM | 1km offshore | - | - | -21.1, 1.5 | 3.6, 2.1 | 3 |
| MPB | Unimpacted | - | - | -25.2, 1.0 | - | 6 |
|  | Impacted | - | - | -21.5, 1.3 | - | 6 |

*values were taken from Harada et al. (2019)



**Table 4**. Bulk $\delta^{13}C$ values, mean $\delta^{13}C$ values of five EAAs (‰) and differences ($\Delta$, ‰) between the two forests in 2017, 20 months after the dieback (mean, SD).

| Group | Taxa | Forest | Bulk $\delta^{13}C$, ‰ | Mean $\delta^{13}C$ of five EAAs, ‰ | n | $\Delta$ Bulk $\delta^{13}C$, ‰ | $\Delta$ Mean $\delta^{13}C$ of five EAAs, ‰ | Permanova p value |
|---|---|---|---|---|---|---|---|---|
| Algal feeder | *Tubuca signata* | Unimpacted | -17.1, 1.4 | -21.9, 1.5 | 3 | 1.7 | 1.4 | 0.90 |
| | | Impacted | -15.4, 1.4 | -20.5, 1.8 | 3 | | | |
| Leaf feeder | *Parasesarma / Episesarma* | Unimpacted | -21.4, 1.5 | -25.3, 1.6 | 3 | 3.1 | 1.4 | 0.40 |
| | | Impacted | -18.3, 0.2 | -23.9, 0.1 | 2 | | | |
| Grazer | *Telescopium telescopium* | Unimpacted | -18.2, 1.9 | -24.0, 1.8 | 3 | 1.5 | 1.8 | 0.80 |
| | | Impacted | -16.7, 1.3 | -22.2, 1.1 | 3 | | | |
| Filter feeder | *Crassostre*a (oyster) | Unimpacted | -19.3, 0.4 | -22.8, 0.2 | 2 | 0.3 | 0.1 | 0.33 |
| | | Impacted | -19.0, 0.8 | -22.9, 0.3 | 2 | | | |
| Mangrove | *Avicennia marina* | Unimpacted | -26.7, 2.2 | -28.8, 0.6 | 2 | 1.3 | 1.8 | 0.67 |
| | | Impacted | -25.4, 0.1 | -27.0, 0.3 | 2 | | | |
| MPB | | Unimpacted | -25.4, 0.8 | -27.4 | 1 | 4.5 | 6.7 | - |
| | | Impacted | -20.9, 1.2 | -20.7 | 1 | | | |





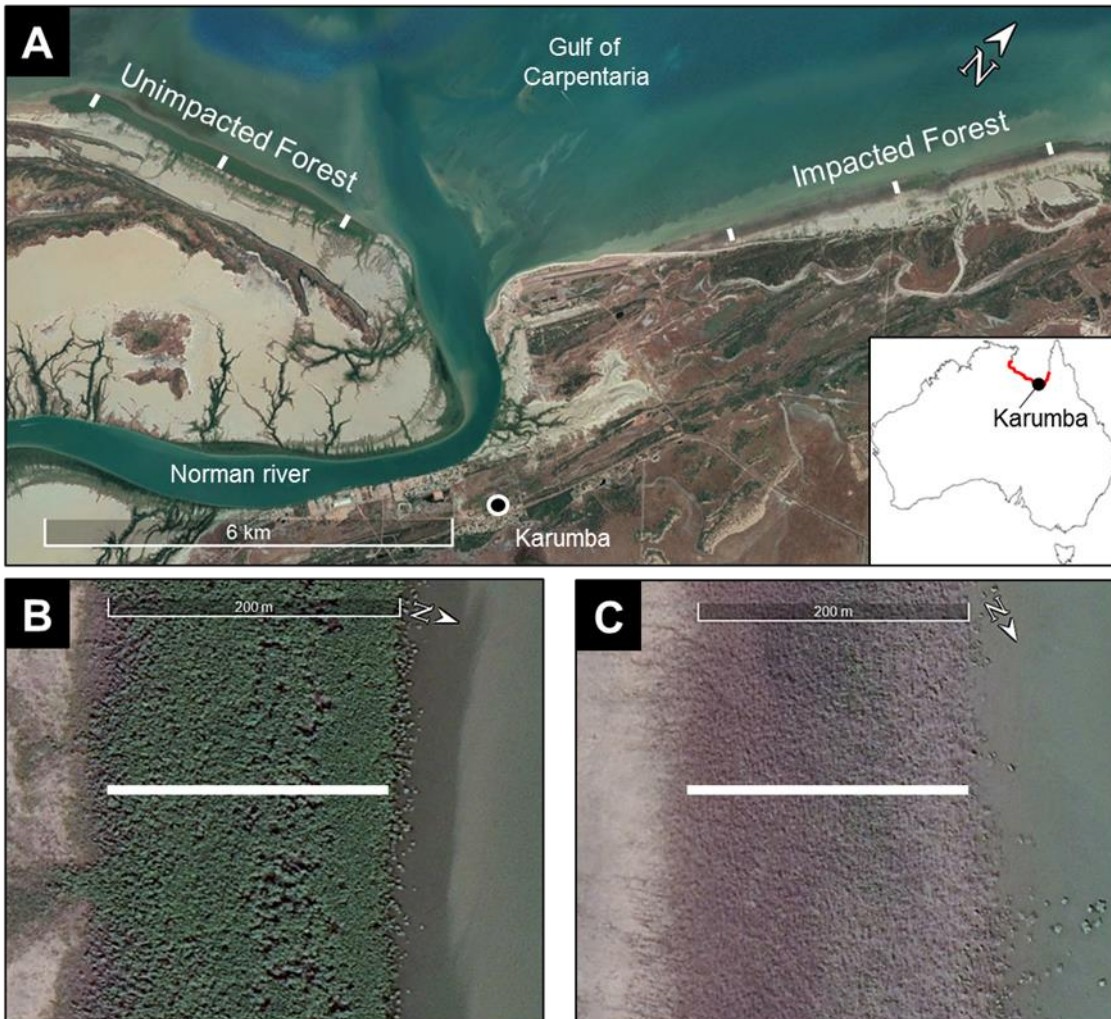

**Figure 1.** The study location at Karumba in the Gulf of Carpentaria, Queensland, Australia (-17.435572S, 140.844766E; image provided by © Google Earth). (A) Three sampling transects within the unimpacted reference site and three within the impacted site (shown as a white line). (B, C) Representative transects from the unimpacted (B) and impacted (C) sites. Each transect was approximately 200m. Samples were collected along each transect from higher to lower intertidal zones. Red indicated mass dieback region.





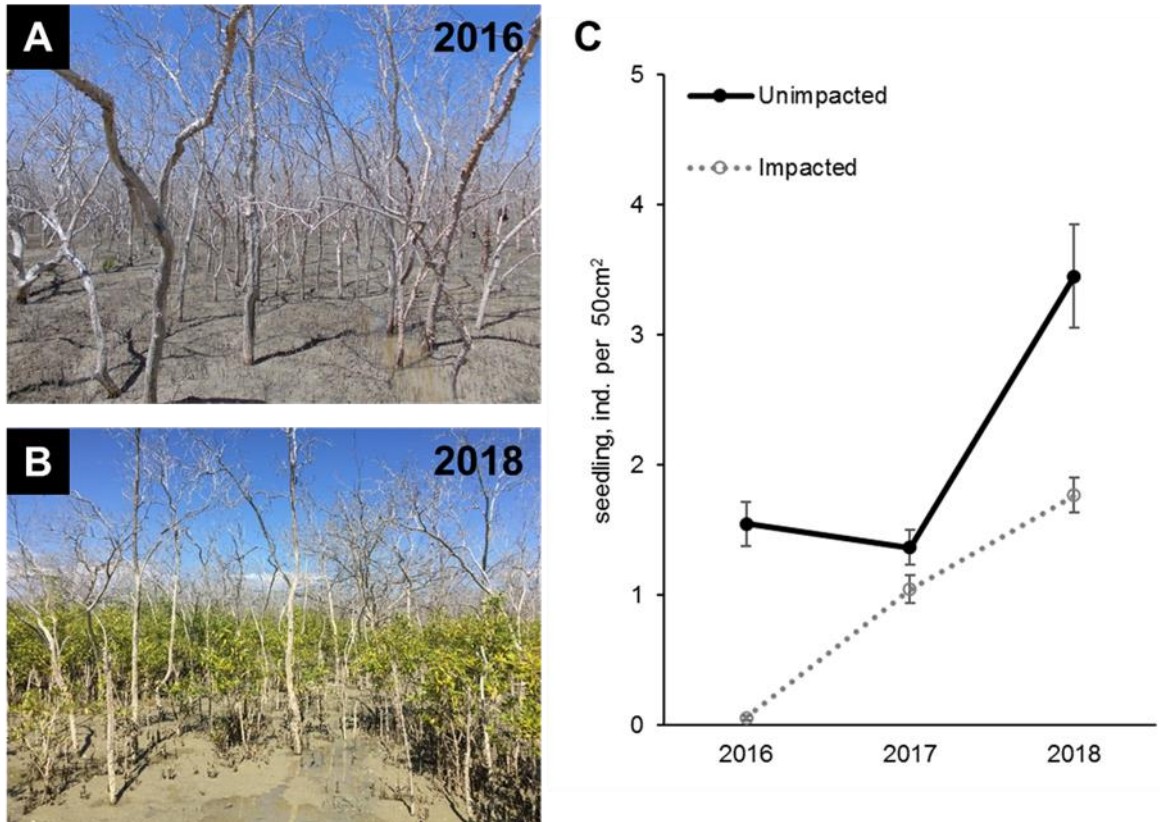

**Figure 2.** Recovery of mangrove vegetation at the impacted site during a two-year period from 2016 to 2018 (A and B; approx. 8 and 32 months, respectively after the dieback event). Seedling and sampling populations of mangrove species (mostly, *A. marina*) significantly increased in the impacted site (C).


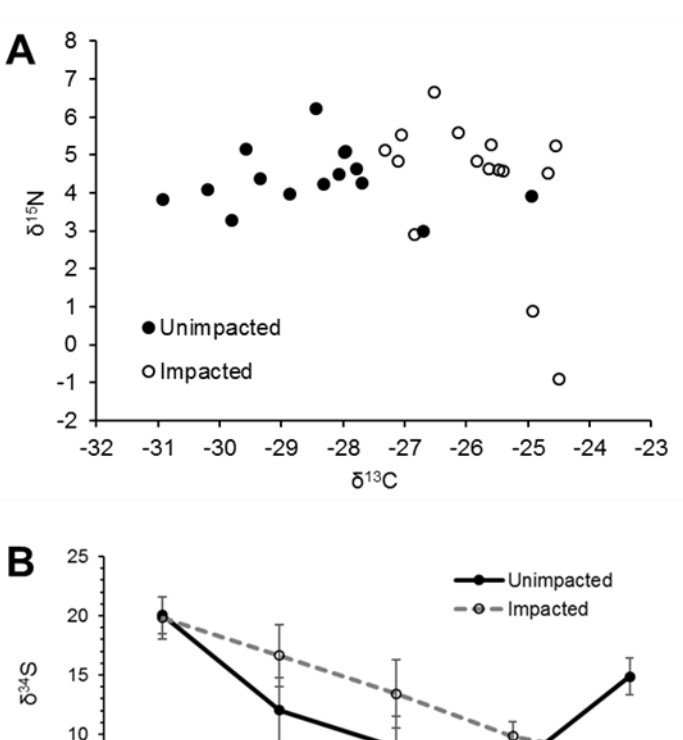

**Figure 3.** CNS isotopic compositions of green leaves of *A. marina* from the unimpacted and impacted sites. All samples were collected in 2018, 32 months after the dieback. (A) Leaf $\delta^{13}C$ and $\delta^{15}N$ values. (B) Leaf $\delta^{34}S$ values across the intertidal zones. Error values are SE.


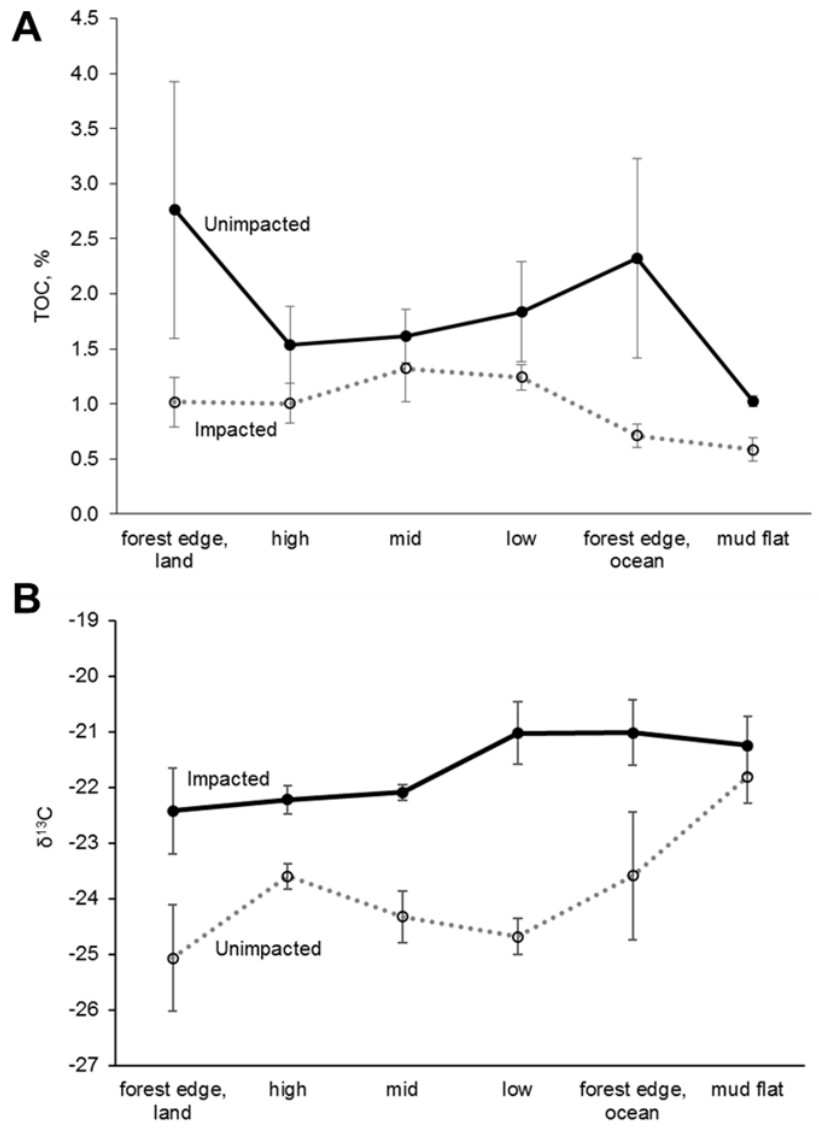

**Figure 4.** C elemental and isotopic compositions of surface (< 0.5 cm) sediment along the unimpacted reference transects vs impacted transects (n=3 per data point). All samples were collected in 2018, 32 months after the dieback. Error values are SE. (A) Sediment TOC, %. (B) Sediment $\delta^{13}C$ values, ‰.


**Figure 5.** Changes in CNS isotopic compositions of mangrove macrofaunal groups with four different feeding modes from 2016, 2017 and 2018 (i.e. 8, 20 and 32 months after the event) between the unimpacted reference and impacted mangrove forest sites. Error bars are ± SE (n=2 to 6 per data point). (*) indicates a significant difference in the year. Mean ± SE values and sample sizes are also provided in Table S1.

**Figure 6.** C isotopic compositions in essential amino acids (EAAs) for four mangrove consumer groups and resources including mangrove leaves (*A. marina*) and MPB from the unimpacted and impacted mangrove sites during 2017 (20 months after the dieback). While there are clear offsets in individual $\delta^{13}C_{EAA}$ values between the two forests, normalized $\delta^{13}C_{EAA}$ fingerprint patterns as per Larsen (2009) shown in Fig S1 did not differ (PERMANOVA p > 0.05, Table 3). Error bars show ± SD.






## Disturbance legacies of the forest dieback

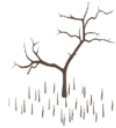

Forest dieback due to extreme climatic events (impacted)
- $\delta^{13}C$ - weak $C_3$ plants signal (due to loss of mangroves)
- $\delta^{15}N$ - Moderate to high (due to degradation and lower N fixation)
- $\delta^{34}S$ - Moderate to high (lower sulfate reduction with a strong seawater sulfate signal)

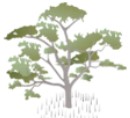

Healthy mangrove ecosystem (background)
- $\delta^{13}C$ - strong $C_3$ plants signal
- $\delta^{15}N$ - Low to moderate (high N fixation)
- $\delta^{34}S$ - Low to moderate (high sulfate reduction)

## Predicted recovery scenarios with isotopic trajectories

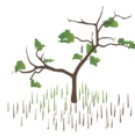

(1) Recovery with no future perturbations
- Environmental conditions allow recolonisation
- Recovery of $\delta^{13}C$, $\delta^{15}N$ and $\delta^{34}S$ to the background

(2) Recovery with future perturbations
- Mangrove recolonisation and recovery of $\delta^{13}C$, $\delta^{15}N$ and $\delta^{34}S$ driven by perturbations e.g. ENSO cycles

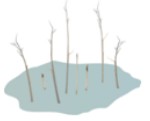

(3) Habitat becomes unsuitable for recolonisation
- Conditions not allow recolonisation e.g. due to extreme climatic events
- Transformed into intertidal mudflats
- No recovery of isotopes

(4) Incomplete recovery
- Reduced habitat size and/or recolonised by other plants such as saltmarshes
- Incomplete recovery of $\delta^{13}C$, $\delta^{15}N$ and $\delta^{34}S$

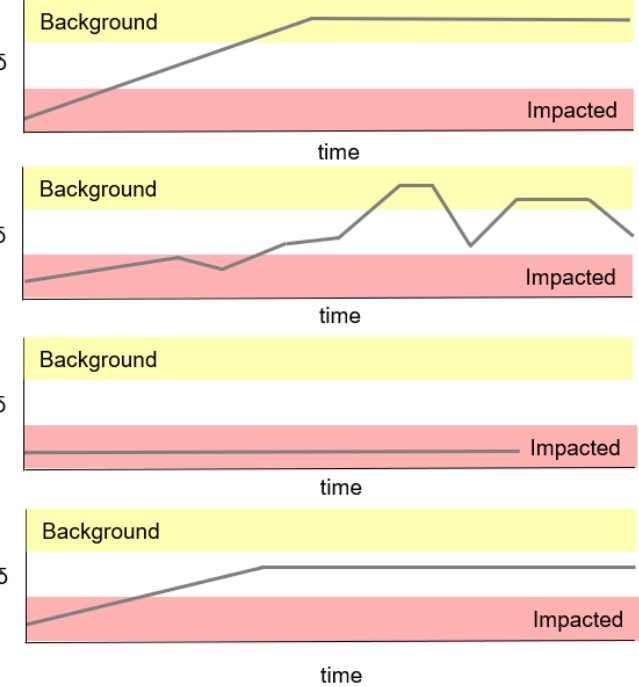

**Figure 7.** A conceptual diagram showing the ecological and biogeochemical legacy of the mangrove forest dieback in the Gulf of Carpentaria and four predicted recovery scenarios of the mangrove ecosystem with isotopic trajectories.