# Peer review of "Stable isotopes track the ecological and biogeochemical legacy of mass mangrove forest dieback in the Gulf of Carpentaria, Australia"

_Biogeosciences, 2019_

## Referee Comment (RC1) · Anonymous Referee #1 · 23 Mar 2020

General comment This study assessed the impact of mangrove dieback and recovery through assessing the changes in vegetation population and biogeochemical variables in the Gulf of Carpentaria. Findings from this study are important to understand the impact of mangrove disturbance on the biogeochemical processes, specifically their interaction between plant and sediment. This study will contribute to the current blue carbon literature while such coastal ecosystems are expected to undergo extreme disturbance in future. The manuscript is well structured and nicely written but can still be improved for some minor correction. Also, I would suggest providing further raw dataset obtained from this study in the supplementary information or via digital data repository platforms such as Mendeley Data and Figshare. Such of these data will provide a bet-

ter understanding for the readers and also be useful for future meta-analysis based study on this topic. The publication of the ms can be recommended after revisions.

Below are my minor comments: Line 15: I would suggest defining the acronym for C, N, S when they first appeared. Sometimes acronyms can make confusion for non-specialist readers. Line 19: Were these samples or applicable for vegetation and sediments only? Line 25: It would be great if data on vegetation population increase are presented in the abstract. Lines 51-55: Most of the cases provided here highlight the impact of mangrove loss. If possible, authors can provide example or reference how mangrove recovery may restore biogeochemical processes. It is important when one of the study aims is to document the ecosystem recovery profile following dieback. Line 100: 'Three field campaigns were carried out in August 2016, 2017 and 2018'. This sentence is redundant with lines 90-91. Line 115: Does this mean that leaves from the impacted site were obtained from seedling rather than survived mature trees? Line 116: I would suggest describing further steps on wood sampling approach, whether samples were done for sapwood only or with heartwood as well? Line 117: It is quite hard to see which stable isotope is applied for each sample. It would be great if the raw data are provided in Supplementary Information or online database. Line 120: In this section, maybe the readers want to know the reason for having a surface (<0.5 cm) and subsurface (0.5-20 cm) sediment samplings. Line 121: 'each forest' do you mean each zone? How many soil core per zone? Line 133: Was number of the sample here denotes the number of photographs or number of quadrats? How many quadrats per forest zone at each transect? Line 191: Was the variation similar to the impacted site? re: 34S depleted from higher to the lower tidal zone Line 259: Double increased? Here may worth to discuss why both unimpacted and impacted sites show similar mangrove seedling increase, despite they have with different number and rates. Line 271: In related to Kelleway et al 2018, was 13C between leaf and wood different significantly from this dieback study? Line 324: 'lower mangrove C inputs' change mangrove with autochthonous? Line 326: 'The surface sediment (0 - 0.5 cm) differed relatively more than the deeper (0.5 to 20 cm) fraction' Sorry, it is quite hard to follow this sentence.

Line 328: How about C/N ratio? It would be great to explore further roles of C/N ratio to support the findings in addition to elemental and isotope variation.

Table 1: Thanks. This table is really helpful to understand the scattered sampling time and what was sampled. Table 2: it is quite unusual to have a comma between mean and SD. I would suggest replacing the comma with $\pm$ here and elsewhere. Figure 2: In the graph, I would suggest providing seedling per hectare instead of per quadrat. Figure 3: Were the authors collect the wood sample as well for SIA? Is there a possibility of presenting 13C and 15N in the same way with 34S, from landward to seaward? Figure 7: It is a nice conceptual figure. Please clarify if isotopes denote for both plant and sediment.

[Figure]

---

## Referee Comment (RC2) · Martin Zimmer (Referee) · 3 Jun 2020

The authors provide data from element and stable isotope analyses in order to better understand post-die-off dynamics of a mangrove ecosystems. They interpret an observed enrichment in heavier isotopes as indicators of reduced C and N fixation and reduced S reduction in the impacted mangrove stand, while the increasing number of mangrove recruits over time suggests recovery of the vegetation. The lack of recovery of CNS cycling after 32 months, by contrast, is considered an indicator for the biogeochemical legacy of the mass mortality event.

Introduction: The praise of the stable isotope approach should certainly also include

some mentioning of its flaws and weaknesses. Among these, the changes in the iso-topic signature are not as globally "predictable" as the first paragraph of the Introduction suggests: many of these changes do not only depend on the species (both consumer and resource) involved but also on the specific environmental conditions... I suggest the first and second paragraph be merged (as they state essentially the same), following a first paragraph of extreme events (currently 2nd paragraph).

Methods: Before learning about the die-back event (and hypotheses on its causes), I would like to get some information about the mangroves themselves, such as species composition, forest structure and so on! It seems Avicennia marina is/was the pre-dominant species in the study area. It is interesting that hypersalinization (as a result of drought) is mentioned as major causative agent of the mass mortality. As A. ma-rina is known to also occur under quite adverse conditions (e.g., at distribution limits of mangroves), wouldn't we assume that it is as tolerant to salinity stress as, e.g., A. germinans from the AEP? It would be nice to get at least an idea of the sediment salinity this hypersalinization resulted in. The reader might also be highly interested in understanding why the mangrove stand north of the river mouth was impacted, while the nearby(!) stand south-west of the river mouth was not. It is obvious that 3 transects were monitored in each of the two stands – how many sampling plots were established in each transect? How were the data from these plots handled (pooled?, ...?)? We need to better understand the (spatial) details of the sampling design! Some more details about the "wood samples" would be helpful: how deep? where on the stem? Etc... According to the hydrodynamics of the area, do the offshore water samples reflect material that is likely to be washed into the mangroves or to be derived from the mangroves? How were the photos taken to allow for relating the number of the seedlings on the photo to a given (unit of) area? Even though the transects were cho-sen as to render the sites for comparison as similar as possible, there remains the fact that "unimpacted" and "impacted" are not replicated – strictly speaking, we are com-parison two sites, one of which is by chance impacted, the other one is not. In this very particular case, I don't consider this a real issue, as the difference is very clear,

but I would like to see that the authors take this non-replicated comparison of two sites that than results in generalized conclusions on "impacted" versus "unimpacted" into account and at least mention this restriction to their conclusions.

Results: "had a 34S value of 16.6‰... $compared to which value for the unimpacted site? l.225 - 230 : these values do not seem to be SIGNIFICNATLY different, though? l.230 ff(and throughout) : what is the "forest type" here? I think we are just comparing one impacted and one unimpacted stand (not two forest types), and I say as above (and throughout) — is "consistently" significant? It doesn't look as if it is (except for 2018)... If the values are not signi — please clarify$!

Very minor linguistics: l.181: "than at the unimpacted site" l.183: "dominant mangroves species, A. marina, did not differ" l.211: "than those from the unimpacted site" l.218: "was similar to value of those collected in the mudflat"

Discussion: "mangrove degradation may be followed by fast colonisation of non-mangrove herbaceous species" – this is an important statement on a general and global problem: in the Caribbean, Acrostichum aureum, the Golden mangrove fern, builds up a dense canopy in disturbed/clear-felled mangrove areas. As this species, as well as congenerics, also occur in the IWP: was the impacted forest (re-)colonized by the fern, or is there no propagule pool available in the vicinity? l.265: why would the "stomatal conductance" be reduced in the impacted site? The environmental conditions were very similar (c.f. Methods), while one site showed mass mortality and the other one did not – what actually is/was the (environmental) difference between these two sites? Why did the mangroves die here but not there? Is the biogeochemical pattern observed a legacy of the die-back, or might it be related to the reason for the die-back (while a nearby mangrove did not exhibit mass mortality)? Several potential reasons for the observed 13C pattern are listed – don't the authors want to discuss these? l.275: what might these "chronic stresses" be? Are they a consequence of the die-back, or are they the reason (the drought that seems to have caused the mass mortality can probably not be considered a "chronic stress" but rather a massive disturbance)? l.289: this is very interesting! I would have expected lower rather than

higher variability in (sediment/microbial) processes upon such string disturbance – can you expand on this to explain how/why the drought and/or die-back would increase the variability of processes? l.315: this interpretation of the findings suggests that at the impacted site it was dead wood that was sampled (from standing dead stems?), whereas wood from living trees was sampled at the unimpacted site – is that correct?

Fauna: before we can go into this discussion, the above issue of whether "consistent"/"substantial" is "significant" needs be clarified. Only IF the values are significantly different, it will make sense to discuss or interpret such differences! l.356: I don't follow this line of argument: Bui & Lee (2014) stress a potential enrichment by up to 5 – here we have a difference of 6-7 . . . is this sufficient to indicate "some additional contributions"? l.363: does that mean that mangrove leaves did not play a role as food source in BOTH forests? If so, this cannot be an effect of the mass mortality, and –of course- we would then not expect any change over time, as this observation would have nothing to do with mangrove recovery after disturbance. . . l.395: I don't understand "can reflect consumer tissues with little isotope effect" – how do the patterns in producers reflect patterns in consumers; shouldn't it be the other way round? l.403: what is it that mostly affect MPB? Besides the biotic changes, we would expect much more light, and thus, higher evaporation and less water at the impacted than at the unimpacted site. This already will change MPB drastically. l.425: I do not understand how you derive these scenarios from the present study? I kind of agree with these potential scenarios (there might be other possibilities), but how does this relate to, how is this justified by, the present study? Minor: l.410: omit "-"

---

## Author Comment (AC1) · 24 Jun 2020

| Reviewer Comment | Author Response |
|---|---|
| **Referee #1** | |
| **General comment** | |
| This study assessed the impact of mangrove dieback and recovery through assessing the changes in vegetation population and biogeochemical variables in the Gulf of Carpentaria. Findings from this study are important to understand the impact of mangrove disturbance on the biogeochemical processes, specifically their interaction between plant and sediment. This study will contribute to the current blue carbon literature while such coastal ecosystems are expected to undergo extreme disturbance in future. The manuscript is well structured and nicely written but can still be improved for some minor correction. Also, I would suggest providing further raw dataset obtained from this study in the supplementary information or via digital data repository platforms such as Mendeley Data and Figshare. Such of these data will provide a better understanding for the readers and also be useful for future meta-analysis based study on this topic. The publication of the ms can be recommended after revisions. | We thank the reviewer for the constructive feedback on the manuscript and will modify it to clarify the points raised. As suggested, we will also provide the entire raw dataset. |
| **Minor comments** | |
| Line 15: I would suggest defining the acronym for C, N, S when they first appeared. Sometimes acronyms can make confusion for non-specialist readers. | We will define the acronym for C, N, S at the first appearance. |
| Line 19: Were these samples or applicable for vegetation and sediments only? | The samples include invertebrates, plants and sediments. We will rewrite the sentence to clarify this. |
| Line 25: It would be great if data on vegetation population increase are presented in the abstract. | We agree. We will provide vegetation data in the abstract. |
| Lines 51-55: Most of the cases provided here highlight the impact of mangrove loss. If possible, authors can provide example or reference how mangrove recovery may restore biogeochemical processes.  It is important when one of the study aims is to document the ecosystem recovery profile following dieback. | Studies that show how mangrove recovery restores biogeochemical process are limited, but we will improve this section by providing references and/or examples. |
| Line 100: 'Three field campaigns were carried out in August 2016, 2017 and 2018'. This sentence is redundant with lines 90-91. | We will remove the sentence (Line 100). |
| Line 115: Does this mean that leaves from the impacted site were obtained from seedling rather than survived mature trees? | Leaves were from regrowth from survived trees. We will rewrite the sentence to clarify this. |
| Line 116: I would suggest describing further steps on wood sampling approach, whether samples were done for sapwood only or with heartwood as well? | Samples were from sapwood. We will add more information on wood sampling. |

| | |
|---|---|
| Line 117: It is quite hard to see which stable isotope is applied for each sample. It would be great if the raw data are provided in Supplementary Information or online database. | We will provide raw data. |
| Line 120: In this section, maybe the readers want to know the reason for having a surface (<0.5 cm) and subsurface (0.5-20 cm) sediment samplings. | The reason for having sediment samples from two depths is to compare surface sediments that represent the recent deposition and micophytobenthos, with the subsurface fraction which represents a long-term average. We will reword the sentence to clarify this. |
| Line 121: 'each forest' do you mean each zone? How many soil core per zone? | Sediment cores were independent samples from the surface sediment. Samples (n=2 per transect) were collected from the mid intertidal zone. We will rewrite the sentence to clarify this. |
| Line 133: Was number of the sample here denotes the number of photographs or number of quadrats? How many quadrats per forest zone at each transect? | A photo was taken for each quadrat, so the number of photos and number of quadrats are the same. The quadrat sampling was carried out at the mid intertidal zone. We will clarify this in the method. |
| Line 191: Was the variation similar to the impacted site? re: 34S depleted from higher to the lower tidal zone | Yes, in both forests, leaf $\delta^{34}$S values decreased from the higher to lower intertidal zones. |
| Line 259: Double increased? Here may worth to discuss why both unimpacted and impacted sites show similar mangrove seedling increase, despite they have with different number and rates. | We agree. We will discuss this. |
| Line 271: In related to Kelleway et al 2018, was 13C between leaf and wood different significantly from this dieback study? | It seems like the wood samples are more enriched than the leaves, but we do not have enough wood samples to make this comparison and also the wood samples were independently sampled from the leaves. |
| Line 324: 'lower mangrove C inputs' change mangrove with autochthonous? | We will change "mangrove" to "autochthonous". |
| Line 326: 'The surface sediment (0 - 0.5 cm) differed relatively more than the deeper (0.5 to 20 cm) fraction' Sorry, it is quite hard to follow this sentence. | We will rewrite the sentence to make it easier to follow. |
| Line 328: How about C/N ratio? It would be great to explore further roles of C/N ratio to support the findings in addition to elemental and isotope variation. | Thank you. We agree. We will explore the C/N ratio data to see if it can support the findings. |
| Table 1: Thanks. This table is really helpful to understand the scattered sampling time and what was sampled.

Table 2: it is quite unusual to have a comma between mean and SD. I would suggest replacing the comma with ± here and elsewhere. | We will use ± instead of comma between mean and SD. |
| Figure 2: In the graph, I would suggest providing seedling per hectare instead of per quadrat. | Thank you, we agree. Since the size of the quadrat is very small compared with a hectare, seedling per m2 will be used in the figure. |
| Figure 3: Were the authors collect the wood sample as well for SIA? Is there a possibility | Wood samples were only collected from the mid intertidal zone, so we can not present the data in the same way. |

| | |
|---|---|
| of presenting 13C and 15N in the same way with 34S, from landward to seaward? | |
| Figure 7: It is a nice conceptual figure. Please clarify if isotopes denote for both plant and sediment. | We will indicate in the figure that the isotopes indicate animals, plants and sediment. |

**Referee #2 Martin Zimmer**

**General comment**

| | |
|---|---|
| The authors provide data from element and stable isotope analyses in order to better understand post-die-off dynamics of a mangrove ecosystems. They interpret an observed enrichment in heavier isotopes as indicators of reduced C and N fixation and reduced S reduction in the impacted mangrove stand, while the increasing number of mangrove recruits over time suggests recovery of the vegetation. The lack of recovery of CNS cycling after 32 months, by contrast, is considered an indicator for the biogeochemical legacy of the mass mortality event. | We thank Dr Martin Zimmer for the constructive feedback on the manuscript. |
| Introduction: The praise of the stable isotope approach should certainly also include some mentioning of its flaws and weaknesses. Among these, the changes in the isotopic signature are not as globally "predictable" as the first paragraph of the Introduction suggests: many of these changes do not only depend on the species (both consumer and resource) involved but also on the specific environmental conditions: : : I suggest the first and second paragraph be merged (as they state essentially the same), following a first paragraph of extreme events (currently 2nd paragraph). | We agree. The flaws and weaknesses of the stable isotope approach will be mentioned in the introduction and included in the interpretation of these results in the discussion. We will merge the first paragraph and the third paragraph to provide one paragraph of the stable isotope approach, following the paragraph of extreme events. |
| Methods: Before learning about the die-back event (and hypotheses on its causes), I would like to get some information about the mangroves themselves, such as species composition, forest structure and so on! It seems Avicennia marina is/was the predominant species in the study area. | We will provide some more information on the characteristics of the mangrove forest studied such as species composition and forest structure before we give information of the die-back event. |
| It is interesting that hypersalinization (as a result of drought) is mentioned as major causative agent of the mass mortality. As A. marina is known to also occur under quite adverse conditions (e.g., at distribution limits of mangroves), wouldn't we assume that it is as tolerant to salinity stress as, e.g., A. germinans from the AEP? It would be nice to get at least an idea of the sediment salinity this hypersalinization resulted in. The reader might also be highly interested in understanding why the mangrove stand north of the river mouth was impacted, while the nearby(!) stand south-west of the river mouth was not. | The cause of this mangrove dieback was reported by Duke et al 2017 and Harris et al 2017 (cited in the present manuscript). It is thought that there were combined effects from drought conditions due to lower rainfalls, in combination with lower sea levels due to large sale climatic patterns (El Nino Southern Oscillation, and Indian Ocean Dipole). There is also a recent paper (Sippo et al. in press) which discusses the cause of the dieback, including: climate data, sediment geochemistry and groundwater availability. We will summarise those studies in more details to give a better idea of the causality. |

| | We can only hypothesize as to why the mangrove stand north of the river mouth was impacted while stand south-west of the river is not. We consider that this maybe due to river influence. It seems that the river outlet turns south-west (Fig 1), so it is likely that the south-west stand has more river influence. Other possibilities include localised groundwater flow paths. |
|---|---|
| It is obvious that 3 transects were monitored in each of the two stands – how many sampling plots were established in each transect? How were the data from these plots handled (pooled?, : : :?)? We need to better understand the (spatial) details of the sampling design! | The number of sampling plots varied among samples. For example, 5 plots for mangrove leaves and 6 plots for sediment along the tidal zone. Data from these plots were pooled. To clarify the spatial details of the sampling design, we will add a table with number of plots for each analysis. |
| Some more details about the "wood samples" would be helpful: how deep? where on the stem? Etc: : : | Wood samples were collected at the mid tidal zone. Sapwood (diameter 5 cm to 15cm) were analysed. We will add more details about the wood sampling. |
| According to the hydrodynamics of the area, do the offshore water samples reflect material that is likely to be washed into the mangroves or to be derived from the mangroves? | The mangrove area is adjacent to an extensive area of mudflats. Material derived from the mangrove area is likely diluted and the offshore water samples mostly reflect material that is likely to be washed into the mangrove such as POM and phytoplankton. We will add a more detailed explanation for this. |
| How were the photos taken to allow for relating the number of the seedlings on the photo to a given (unit of) area? | For each photo, a 50cm x 50cm of quadrat was used to indicate a unit of area. These details will be added |
| Even though the transects were chosen as to render the sites for comparison as similar as possible, there remains the fact that "unimpacted" and "impacted" are not replicated – strictly speaking, we are comparison two sites, one of which is by chance impacted, the other one is not. In this very particular case, I don't consider this a real issue, as the difference is very clear, but I would like to see that the authors take this non-replicated comparison of two sites that than results in generalized conclusions on "impacted" versus "unimpacted" into account and at least mention this restriction to their conclusions. | We agree. We will mention this restriction to the conclusion. |
| Results: "had a 34S value of 16.6‰. . . compared to which value for the unimpacted site? | Wood samples for the unimpacted site did not have enough S to determine the isotope values, therefore we do not have sufficient data to make this comparison. |
| l:225 - 230 : these values do not seem to be SIGNIFICNATLY different; though? | Figure 5 shows the ANOVA results and which samples significantly differed, but we will rewrite the sentence to clarify this. |
| l:230 ff(and throughout) : what is the "forest type" here? I think we are just comparing one impacted and one unimpacted stand (not two forest types); and I suggest to stick to this (like above)! | We will use impacted and unimpacted throughout the ms. |
| l:236 as above (and throughout ) - - is "consistently" significant? It doesn't look as if it is(except for 2018). . . If the values are not | We will indicate which means are significantly different in the text. |

| | |
|---|---|
| significantly different ; we cannot consider them" different"; - please clarify! | |
| Very minor linguistics:
l.181: "than at the unimpacted site"
l.183: "dominant mangroves species, A. marina, did not differ"
l.211: "than those from the unimpacted site" l.218: "was similar to value of those collected in the mudflat" | Thank you. We will correct the linguistic errors. |
| Discussion: "mangrove degradation may be followed by fast colonisation of nonmangrove herbaceous species" – this is an important statement on a general and global problem: in the Caribbean, Acrostichum aureum, the Golden mangrove fern, builds up a dense canopy in disturbed/clear-felled mangrove areas. As this species, as well as congenerics, also occur in the IWP: was the impacted forest (re-)colonized
by the fern, or is there no propagule pool available in the vicinity? | The impacted site was not colonized by the fern. There was fast colonisation by mangroves, so it is likely that a propagule pool is available in the vicinity. |
| l.265: why would the "stomatal conductance" be reduced in the impacted site? The environmental conditions were very similar (c.f. Methods), while one site showed mass mortality and
the other one did not – what actually is/was the (environmental) difference between these two sites? Why did the mangroves die here but not there? Is the biogeochemical pattern observed a legacy of the die-back, or might it be related to the reason for the die-back (while a nearby mangrove did not exhibit mass mortality)? Several potential reasons for the observed 13C pattern are listed – don't the authors want to discuss these? | There is less canopy cover at the impacted site, so there could be higher evaporation and lower water availability, which can reduce stomatal conductance.

Leaves were depleted in $^{13}$C at the unimpacted site, suggesting that there could be higher water availability at the unimpacted site. Several potential reasons for the observed $^{13}$C pattern will be discussed in more detail. |
| l.275: what might these "chronic stresses" be? Are they a consequence of the die-back, or are they the reason (the drought that seems to have caused the mass mortality can probably not be considered a "chronic stress" but rather a massive disturbance)? | Such environmental stresses may include hypersalinization of sediments and hydric, thermal and radiant stresses following mangrove losses (e.g. canopy loss). This is mentioned in l.276-7 |
| l.289: this is very interesting! I would have expected lower rather than higher variability in (sediment/microbial) processes upon such string disturbance – can you expand on this to explain how/why the drought and/or die-back would increase the variability of processes? | We will expand on this. It is possible that the disturbance caused patchiness. The disturbed system may be at more unstable conditions and changing. |
| l.315: this interpretation of the findings suggests that at the impacted site it was dead wood that was sampled (from standing dead stems?),
whereas wood from living trees was sampled at the unimpacted site – is that correct? | Yes, we sampled dead wood from the impacted site and living wood from the unimpacted site. We will mention this in the ms. |
| Fauna: before we can go into this discussion, the above issue of whether "consistent"/"substantial" is "significant" needs be clarified. Only IF the values | We agree. We will use "significant" to indicate which samples statistically differed. |

| | |
|---|---|
| are significantly different, it will make sense to discuss or interpret such differences! | |
| l.356: I don't follow this line of argument: Bui & Lee (2014) stress a potential enrichment by up to 5 – here we have a difference of 6-7 : : : is this sufficient to indicate "some additional contributions"? | Bui & Lee (2014) fed crabs with mangrove leaves. The crabs displayed an enrichment of about 5‰ from the leaves, so that the difference of 6-7‰ most likely indicates there was also a more enriched source. We will rewrite the line to clarify this. |
| l.363: does that mean that mangrove leaves did not play a role as food source in BOTH forests? If so, this cannot be an effect of the mass mortality, and – of course we would then not expect any change over time, as this observation would have nothing to do with mangrove recovery after disturbance: : | We will add more information to clarify this. We consider that mangrove leaves played a minor role as food source, but other sources such as phytoplankton and MPB played a more important role in both forests. However, the presence or absence of mangroves can still change the isotope values of consumers, consistent with the finding for other studies, e.g., Bernardino et al. (2018). |
| l.395: I don't understand "can reflect consumer tissues with little isotope effect" – how do the patterns in producers reflect patterns in consumers; shouldn't it be the other way round? | Isotopic compositions in essential amino acids can be reflected in the consumers with little trophic isotopic fractionation. We will rewrite the line. |
| l.403: what is it that mostly affect MPB? Besides the biotic changes, we would expect much more light, and thus, higher evaporation and less water at the impacted than at the unimpacted site. This already will change MPB drastically. | We will discuss this in more details. Source of carbon and isotope fractionation can affect the isotope value of MPB. Changes to abiotic factors such as light, evaporation and water availability due to the canopy loss can change both C sources and fractionation. It is thought that lower respiratory input and lower dissolved inorganic C availability could change MPB drastically. |
| l.425: I do not understand how you derive these scenarios from the present study? I kind of agree with these potential scenarios (there might be other possibilities), but how does this relate to, how is this justified by, the present study? | These are likely scenarios and there might be other possibilities. What we have learned from this study is that biochemical changes can be reflected in the isotopic values of organisms. Multi-annual sampling can be used to track their changes overtime and such isotopic information can be used to monitor biogeochemical changes in the future. It can be expected from this study that when the impacted forest is fully recovered, it would be isotopically similar to the unimpacted site. If the forest is unable to recover this may not be observed. |
| Minor: l.410: omit "-" | We will omit "-" |

---

## Author Comment (AC2) · 24 Jun 2020

Please find our detailed replies in the attached PDF.

Please also note the supplement to this comment:
https://www.biogeosciences-discuss.net/bg-2019-468/bg-2019-468-AC2-supplement.pdf

———————————————

---

## Author Response (AR2)

| Reviewer Comment | Author Response | New location |
|---|---|---|
| **Referee #1** | | |
| **General comment** | | |
| This study assessed the impact of mangrove dieback and recovery through assessing the changes in vegetation population and biogeochemical variables in the Gulf of Carpentaria. Findings from this study are important to understand the impact of mangrove disturbance on the biogeochemical processes, specifically their interaction between plant and sediment. This study will contribute to the current blue carbon literature while such coastal ecosystems are expected to undergo extreme disturbance in future. The manuscript is well structured and nicely written but can still be improved for some minor correction. Also, I would suggest providing further raw dataset obtained from this study in the supplementary information or via digital data repository platforms such as Mendeley Data and Figshare. Such of these data will provide a better understanding for the readers and also be useful for future meta-analysis based study on this topic. The publication of the ms can be recommended after revisions. | We thank the reviewer for the constructive feedback on the manuscript and have now modified it to clarify the points raised.

As suggested, we have now provided additional dataset in the supplementary information for a better understanding for the readers. Table S1, S2, S3 and S5. | |
| **Minor comments** | | |
| Line 15: I would suggest defining the acronym for C, N, S when they first appeared. Sometimes acronyms can make confusion for non-specialist readers. | We have now defined the acronym for C, N, S at the first appearance. | Ln 16 |
| Line 19: Were these samples or applicable for vegetation and sediments only? | The samples include invertebrates, plants and sediments. We have now changed the sentence to clarify this (blue is new text).

"Invertebrates and associated organic matter including mangroves, and sediments from the impacted ecosystem showed enrichment in $^{13}C$, $^{15}N$ and $^{34}S$ relative to those from an adjacent unimpacted reference ecosystem…" | Ln 19-21 |
| Line 25: It would be great if data on vegetation population increase are presented in the abstract. | We agree. We have now provided vegetation data in the abstract. "The seedling density increased from 0.2 per $m^2$ in 2016 to 7.1 per $m^2$ in 2018 in the impacted forest". | Ln 25-26 |
| Lines 51-55: Most of the cases provided here highlight the impact of mangrove loss. If possible, authors can provide example or reference how mangrove recovery may restore biogeochemical processes. It is important when one of the study aims is to | We have inserted new sentences and references to improve this section.

"Although mangroves can recover from mortality events, the rate of recovery can be slow. For example, a study of mangrove | Ln 45-48 |

| | | |
|---|---|---|
| document the ecosystem recovery profile following dieback. | mortality attributed to an oil spill incident shows full recovery may take over 50 years (Connolly et al. 2020) and full recovery of belowground C and N stocks after mangrove replantation may take over 40 years (Adame et al., 2018)". | |
| Line 100: 'Three field campaigns were carried out in August 2016, 2017 and 2018'. This sentence is redundant with lines 90-91. | We have now removed the sentence. | |
| Line 115: Does this mean that leaves from the impacted site were obtained from seedling rather than survived mature trees? | Leaves were from regrowth from survived trees. We have now changed the sentence to clarify this. "In the impacted site, leaves were collected from regrowth of trees that had survived" | Ln 124 - 125 |
| Line 116: I would suggest describing further steps on wood sampling approach, whether samples were done for sapwood only or with heartwood as well? | Samples were from sapwood. We have now added more information on wood sampling. "wood samples (n=2, 5 to 25cm diameter) were collected using a hand saw from stems at chest height from the mid intertidal zone of each forest. Dead trees were sampled at the impacted site. Two to three bulk SIA measurements were made from sapwood (2 to 3cm deep) of each sample and measurements were averaged." | Ln 126-128 |
| Line 117: It is quite hard to see which stable isotope is applied for each sample. It would be great if the raw data are provided in Supplementary Information or online database. | We have now provided additional data in the supplementary information (Table S1, S2, S3 and S5). | |
| Line 120: In this section, maybe the readers want to know the reason for having a surface (<0.5 cm) and subsurface (0.5-20 cm) sediment samplings. | The reason for having sediment samples from two depths is to compare surface sediments that represent the recent deposition and microphytobenthos, with the subsurface fraction which represents a long-term average. We have now reworded the sentence to clarify this.

"In 2018, surface (<0.5 cm) sediments that represent the recent deposition and microphytobenthos (MPB) were collected along each transect. Additionally, subsurface (0.5 to 20 cm) sediment samples (n=6) that represent long-term averages were collected at the mid intertidal zone of each forest using a core sampler (5 cm in diameter and 20 cm deep)". | Ln 131-133 |
| Line 121: 'each forest' do you mean each zone? How many soil core per zone? | Sediment cores were independent samples from the surface sediment. Samples (n=2 per transect) were collected from the mid intertidal zone. We have now changed the sentence to clarify this. "Additionally, subsurface (0.5 to 20 cm) sediment samples | Ln 132-133 |

| | (n=6) that represent long-term averages were collected at the mid intertidal zone of each forest using a core sampler (5 cm in diameter and 20 cm deep)". | |
|---|---|---|
| Line 133: Was number of the sample here denotes the number of photographs or number of quadrats? How many quadrats per forest zone at each transect? | A photo was taken for each quadrat, so the number of photos and number of quadrats are the same.

We have now inserted " To estimate mangrove seedling/sapling densities (ind. m$^{-2}$) from each forest and their changes over time, seedling/saplings were counted with a 50 x 50 cm quadrat at the mid intertidal zone. A photo was taken of each quadrat (for 2016, n=124 for the unimpacted forest and n=143 for the impacted forest, for 2017, n=161 and n=175, and for 2018, n=80 and n=117, respectively) and then counts of seedlings and samplings were made in the laboratory". | Ln 143-146 |
| Line 191: Was the variation similar to the impacted site? re: 34S depleted from higher to the lower tidal zone | Yes, in both forests, leaf $\delta^{34}$S values decreased from the higher to lower intertidal zones. This is shown in the Fig 3 and the data is provided in the supplementary information. | Fig 3 |
| Line 259: Double increased? Here may worth to discuss why both unimpacted and impacted sites show similar mangrove seedling increase, despite they have with different number and rates. | We have now discussed this in more detail.

" In both mangrove forests at the Gulf of Carpentaria site, the density of mangrove seedling/samplings significantly increased throughout the period from 2016 to 2018, suggesting that recovery was starting to occur in some areas within 32 months after the dieback and propagule pool was available in the vicinity. The increase in seedling/sampling density at the unimpacted site was unexpected, but this indicates that there was some stress at the unimpacted site during the dieback period and/or the temporal variability of seedling/samplings density was high at the site.". | Ln 264-269 |
| Line 271: In related to Kelleway et al 2018, was 13C between leaf and wood different significantly from this dieback study? | It seems like the wood samples are more enriched than the leaves, but we do not have enough wood samples to make this comparison and also the wood samples were independently sampled from the leaves. | |
| Line 324: 'lower mangrove C inputs' change mangrove with autochthonous? | We have now changed "mangrove" to "autochthonous". | Ln 329 |
| 8Line 326: 'The surface sediment (0 - 0.5 cm) differed relatively more than the deeper (0.5 to 20 cm) fraction' Sorry, it is quite hard to follow this sentence. | We have now changed the sentence. "The surface sediment varied more than the subsurface fraction". | Ln 331 |
| Line 328: How about C/N ratio? It would be great to explore further roles of C/N ratio | Thank you. We have now provided the C/N ratio data in Table 2. The result was discussed in the text. "Despite the substantial variation | Ln 215-216 |

| | | |
|---|---|---|
| to support the findings in addition to elemental and isotope variation. | in TOC and TN, the C/N ratio did not differ significantly between the two sites (ANOVA P > 0.05)". | |
| Table 1: Thanks. This table is really helpful to understand the scattered sampling time and what was sampled.

Table 2: it is quite unusual to have a comma between mean and SD. I would suggest replacing the comma with ± here and elsewhere. | We have now used ± instead of comma between mean and SD. | Table 2
Table 3
Table 4 |
| Figure 2: In the graph, I would suggest providing seedling per hectare instead of per quadrat. | Thank you, we agree. Since the size of the quadrat is very small compared with a hectare, we have now used seedling per m2 in the figure. | Figure 2 |
| Figure 3: Were the authors collect the wood sample as well for SIA? Is there a possibility of presenting 13C and 15N in the same way with 34S, from landward to seaward? | Wood samples were only collected from the mid intertidal zone, so we cannot present the data in the same way. | |
| Figure 7: It is a nice conceptual figure. Please clarify if isotopes denote for both plant and sediment. | We have now indicated this in the figure "δ represents the isotope values of animals, plants and sediment ". | Figure 7 |
| **Referee #2 Martin Zimmer** | | |
| **General comment** | | |
| The authors provide data from element and stable isotope analyses in order to better understand post-die-off dynamics of a mangrove ecosystems. They interpret an observed enrichment in heavier isotopes as indicators of reduced C and N fixation and reduced S reduction in the impacted mangrove stand, while the increasing number of mangrove recruits over time suggests recovery of the vegetation. The lack of recovery of CNS cycling after 32 months, by contrast, is considered an indicator for the biogeochemical legacy of the mass mortality event. | We thank Dr Martin Zimmer for the constructive feedback on the manuscript. | |
| Introduction: The praise of the stable isotope approach should certainly also include some mentioning of its flaws and weaknesses. Among these, the changes in the isotopic signature are not as globally "predictable" as the first paragraph of the Introduction suggests: many of these changes do not only depend on the species (both consumer and resource) involved but also on the specific environmental conditions: : : I suggest the first and second paragraph be merged (as they state essentially the same), following a first paragraph of extreme events (currently 2nd paragraph). | We agree. We have now mentioned the flaws and weaknesses of the stable isotope approach in the introduction and in the interpretation of these results in the discussion. We have now merged also the first paragraph and the third paragraph to provide one paragraph of the stable isotope approach, following the paragraph of extreme events.

"Stable isotope analysis (SIA) provides biogeochemical process information integrated over time and is useful for environmental assessment and monitoring. As elements such as carbon (C), nitrogen (N) and | Ln 50-67 |

| | | |
|---|---|---|
| | sulfur (S) circulate in the biosphere, stable isotopic compositions of $^{13}C/^{12}C$, $^{15}N/^{14}N$ and $^{34}S/^{32}S$ can change in predictable ways due to mixing and fractionation, giving insights into sources and cycling of these elements (Fry 2006). SIA has been widely used in mangrove ecosystem studies to better understand food web interactions (Bouillon et al., 2008; Larsen et al., 2012; Bui and Lee, 2014; Abrantes et al., 2015), nutrient uptake (McKee et al., 2002), water use (Santini et al., 2015; Hayes et al., 2019), cycling of C (Maher et al., 2013a; Maher et al., 2017; Sasmito et al., 2020), N (Fry and Cormier, 2011), S (Raven et al. 2019), and greenhouse gas emissions (Maher et al., 2013b). While traditional field methods such as measuring species composition to evaluate structure and functioning of ecosystems can be time-consuming and expensive, SIA of ecosystem components can evaluate functional aspects of element cycling and food webs in a cost-effective way (Fry, 2006). To quantify food web dynamics, SIA of total organic matter (''bulk'') requires determination of the baseline isotope values of the food web, but this is difficult to achieve, particularly for complex detrital food webs. Some of the uncertainties associated with bulk SIA have been clarified by compound-specific isotope analysis of amino acids (CSIA-AA). This technique has been increasingly used to assess pathways of energy transfer throughout food webs by distinguishing the effects of baseline isotope values and trophic transfer. Distinct isotopic fractionation between two groups of amino acids occurs with each trophic transfer. In general, non-essential and/or trophic AAs show large isotopic fractionation per trophic step, while essential and/or source AAs show little fractionation, reflecting the baseline isotope values of the food web (Ishikawa et al., 2018; Larsen et al., 2013; Ohkouchi et al., 2017)." | |
| Methods: Before learning about the die-back event (and hypotheses on its causes), I would like to get some information about the mangroves themselves, such as species composition, forest structure and so on! It seems Avicennia marina is/was the predominant species in the study area. | We have now provided some more information on the characteristics of the mangrove forest studied before providing the information of the die-back event.

"The Gulf of Carpentaria in tropical Northern Australia is an extensive, shallow coastal gulf. The area mainly consists of low-lying | Ln 79-87 |

| | wetlands and is largely inaccessible with little direct human activity. Mangroves are abundant in the area, but the dry climate limits the extent, diversity, and height of mangroves in the region (Asbridge et al., 2016). The wide tidal wetlands spread along the shoreline with high intertidal saltpans and saltmarsh covering more area than mangroves. Mangroves are often distributed in the seaward margin, typically as a narrow strip and fronted by extensive, shallow mudflats. The distribution of mangroves in this region is associated with tidal and freshwater inundation, river discharge and regular sediment supply through freshwater input. Increased amounts of rainfall and associated flooding and sea level rise were responsible for recent mangrove extension in this region between 1987 and 2014 (Asbridge et al., 2016). *Avicennia marina* was the dominant mangrove species at the study site in Karumba, Gulf of Carpentaria (Fig 1A)". | |
|---|---|---|
| It is interesting that hypersalinization (as a result of drought) is mentioned as major causative agent of the mass mortality. As A. marina is known to also occur under quite adverse conditions (e.g., at distribution limits of mangroves), wouldn't we assume that it is as tolerant to salinity stress as, e.g., A. germinans from the AEP? It would be nice to get at least an idea of the sediment salinity this hypersalinization resulted in. The reader might also be highly interested in understanding why the mangrove stand north of the river mouth was impacted, while the nearby(!) stand south-west of the river mouth was not. | We have now discussed the cause of dieback in more detail.

"The dieback coincided with a weak monsoon (low rainfall), combined with high vapor pressure deficit, and El Niño–Southern Oscillation-induced low sea-levels (Duke et al., 2017; Lovelock et al., 2017; Harris et al., 2017). The drought conditions most likely caused accumulative hydric, thermal and radiant stresses (Duke et al., 2017). In addition to low water availability, iron (Fe) toxicity due to a rapid mobilisation of sedimentary Fe and regional variability in groundwater flows may have also played a role in the dieback (Sippo et al., 2020a)".

We can only hypothesize as to why the mangrove stand north of the river mouth was impacted while stand south-west of the river is not. We have now inserted " Some local factors (e.g. river influence and localized groundwater flow paths) may have kept some mangroves from dying back in the region" | Ln 95-99

Ln 104-105 |
| It is obvious that 3 transects were monitored in each of the two stands – how many sampling plots were established in each transect? How were the data from these plots handled (pooled?, : : :?)? We need to better | The number of sampling plots varied among samples. For example, 5 plots for mangrove leaves and 6 plots for sediment along the tidal zone. Data from these plots were pooled. To clarify the spatial details of the sampling | Table 1 |

| | | |
|---|---|---|
| understand the (spatial) details of the sampling design! | design, we have now added a table with number of plots for each analysis. | |
| Some more details about the "wood samples" would be helpful: how deep? where on the stem? Etc: : : | We have now added more detail about the wood samples.

"wood samples (n=2, 5 to 25cm diameter) were collected using a hand saw from stems at chest height from the mid intertidal zone of each forest. Dead trees were sampled at the impacted site. Two to three bulk SIA measurements were made from sapwood (2 to 3cm deep) of each sample and measurements were averaged." | Ln 126-128 |
| According to the hydrodynamics of the area, do the offshore water samples reflect material that is likely to be washed into the mangroves or to be derived from the mangroves? | The mangrove area is adjacent to an extensive area of mudflats. Material derived from the mangrove area is likely diluted and the offshore water samples mostly reflect material that is likely to be washed into the mangrove such as POM and phytoplankton.

We have now inserted "...particulate organic matter (POM), material such as phytoplankton that is transported to the mangrove". | Ln 140-141 |
| How were the photos taken to allow for relating the number of the seedlings on the photo to a given (unit of) area? | For each photo, a 50cm x 50cm of quadrat was used to indicate a unit of area.

" To estimate mangrove seedling/sapling densities (ind. m$^{-2}$) from each forest and their changes over time, seedling/saplings were counted with a 50 x 50 cm quadrat at the mid intertidal zone. A photo was taken of each quadrat (for 2016, n=124 for the unimpacted forest and n=143 for the impacted forest, for 2017, n=161 and n=175, and for 2018, n=80 and n=117, respectively) and then counts of seedlings and samplings were made in the laboratory." | Ln 143-146 |
| Even though the transects were chosen as to render the sites for comparison as similar as possible, there remains the fact that "unimpacted" and "impacted" are not replicated – strictly speaking, we are comparison two sites, one of which is by chance impacted, the other one is not. In this very particular case, I don't consider this a real issue, as the difference is very clear, but I would like to see that the authors take this non-replicated comparison of two sites that than results in generalized conclusions on "impacted" versus "unimpacted" into account | We agree. We have now mentioned this in the conclusion. "Although the unimpacted and impacted forests were not replicated in this study, the difference between the two sites was clear" | Ln 402-403 |

| | | |
|---|---|---|
| and at least mention this restriction to their conclusions. | | |
| Results: "had a 34S value of 16.6‰. . . compared to which value for the unimpacted site? | Wood samples for the unimpacted site did not have enough S to determine the isotope values, therefore we do not have sufficient data to make this comparison and have now removed the wood 34S values to avoid confusion. | |
| l:225 - 230 : these values do not seem to be SIGNIFICNATLY different; though? | Figure 5 shows the ANOVA results and which samples significantly differed. | |
| l:230 ff(and throughout) : what is the "forest type" here? I think we are just comparing one impacted and one unimpacted stand (not two forest types); and I suggest to stick to this (like above)! | We have now used impacted and unimpacted throughout the ms. | |
| l:236 as above (and throughout ) - - is "consistently" significant? It doesn't look as if it is(except for 2018). . . If the values are not significantly different ; we cannot consider them" different"; - please clarify! | We have now modified the line and indicated which means are significantly different in the text.

"In 2018, leaf feeder $\delta^{13}$C values, grazer $\delta^{34}$S values and algae feeder $\delta^{34}$S values significantly differed between the impacted and unimpacted sites (Turkey post hoc test, P < 0.05), showing no recovery of the invertebrate fauna $\delta^{13}$C and $\delta^{34}$S status 32 months after the dieback event. However, $\delta^{15}$N values became similar between the two forests in 2018 (Fig. 5)". | Ln 243-246 |
| Very minor linguistics:
l.181: "than at the unimpacted site"
l.183: "dominant mangroves species, A. marina, did not differ"
l.211: "than those from the unimpacted site"
l.218: "was similar to value of those collected in the mudflat" | Thank you. We have now corrected the linguistic errors. | |
| Discussion: "mangrove degradation may be followed by fast colonisation of nonmangrove herbaceous species" – this is an important statement on a general and global problem: in the Caribbean, Acrostichum aureum, the Golden mangrove fern, builds up a dense canopy in disturbed/clear-felled mangrove areas. As this species, as well as congenerics, also occur in the IWP: was the impacted forest (re-)colonized by the fern, or is there no propagule pool available in the vicinity? | The impacted site was not colonized by the fern. There was fast colonisation by mangroves, so it is likely that a propagule pool was available in the vicinity.

We have now inserted "Further, mangrove degradation may be followed by fast colonisation of non-mangrove herbaceous species, e.g. succulent saltmarsh (Mbense et al., 2016; McKee et al., 2007; Rashid et al., 2009). However, this was not largely evident from our study site. In both mangrove forests at the Gulf of Carpentaria site, the density of mangrove seedling/samplings significantly increased throughout the period from 2016 to 2018, suggesting that recovery was starting to occur in some areas within 32 months after | Ln 262-267 |

| | the dieback and propagule pool was available in the vicinity". | |
|---|---|---|
| l.265: why would the "stomatal conductance" be reduced in the impacted site? The environmental conditions were very similar (c.f. Methods), while one site showed mass mortality and the other one did not – what actually is/was the (environmental) difference between these two sites? Why did the mangroves die here but not there? Is the biogeochemical pattern observed a legacy of the die-back, or might it be related to the reason for the die-back (while a nearby mangrove did not exhibit mass mortality)? Several potential reasons for the observed 13C pattern are listed – don't the authors want to discuss these? | We have now discussed several potential reasons for the observed $^{13}C$ pattern in more detail.

"Overall, $^{13}C$-enriched leaf $\delta^{13}C$ values in the impacted forest likely suggest that there are chronic stresses associated with the dieback event that reduced stomatal conductance. Such environmental stresses may include hypersalinization of sediments and hydric, thermal and radiant stresses following canopy losses that cause higher evaporation and lower water availability. The leaves at the unimpacted site were largely depleted in $^{13}C$, suggesting that there was higher water availability at the unimpacted site, possibly associated with regional variability in groundwater flows, e.g., Sippo et al. (2020)". | Ln 284-289 |
| l.275: what might these "chronic stresses" be? Are they a consequence of the die-back, or are they the reason (the drought that seems to have caused the mass mortality can probably not be considered a "chronic stress" but rather a massive disturbance)? | We have now inserted "Such environmental stresses may include hypersalinization of sediments and hydric, thermal and radiant stresses following canopy losses that cause higher evaporation and lower water availability". | Ln 285-289 |
| l.289: this is very interesting! I would have expected lower rather than higher variability in (sediment/microbial) processes upon such string disturbance – can you expand on this to explain how/why the drought and/or die-back would increase the variability of processes? | We have now inserted "The higher variability in leaf $\delta^{15}N$ in the impacted forest suggests higher variability in processes affecting the $\delta^{15}N$ status of available N. For example, changes to sediment conditions including redox transitions and soil moisture content following the dieback may have affected microbial processes of N, whereas the unimpacted forest may have had more stable sediment conditions and N processes". | Ln 299-302 |
| l.315: this interpretation of the findings suggests that at the impacted site it was dead wood that was sampled (from standing dead stems?), whereas wood from living trees was sampled at the unimpacted site – is that correct? | Yes, we sampled dead wood from the impacted site and living wood from the unimpacted site. We have now mentioned this in the ms. Since we do not have sufficient S isotope data from the two forests for comparisons, we have deleted S isotope data and also the interpretation to simplify the ms and avoid confusions. | |
| Fauna: before we can go into this discussion, the above issue of whether "consistent"/"substantial" is "significant" needs be clarified. Only IF the values are significantly different, it will make sense to discuss or interpret such differences! | We have now used "significant" to indicate which samples statistically differed. | |
| l.356: I don't follow this line of argument: Bui & Lee (2014) stress a potential enrichment by up to 5 – here we have a | We have modified the lines to clarify this. | |

| | | |
|---|---|---|
| difference of 6-7 : : : is this sufficient to indicate "some additional contributions"? | "Typical mangrove leaf-eating sesarmid crab species generally have tissue $\delta^{13}C$ values within about +5‰ from mangrove detritus (Bui and Lee, 2014)". | Ln 352-353 |
| | "The leaf feeders were relatively depleted in $^{13}C$, with $\delta^{13}C$ values of about -21 to -18‰, likely due to some use of mangrove leaves". | Ln 358-359 |
| l.363: does that mean that mangrove leaves did not play a role as food source in BOTH forests? If so, this cannot be an effect of the mass mortality, and –of course we would then not expect any change over time, as this observation would have nothing to do with mangrove recovery after disturbance: : : | We consider that mangrove leaves played a minor role as food source, but MPB played a more important role in both forests. However, the presence or absence of mangroves can still change the isotope values of consumers, consistent with the finding for other studies, e.g., Bernardino et al. (2018). We have mentioned this in the text. "These findings did not support changes to feeding dependency following mangrove loss but suggested that the overall differences in the consumer bulk $\delta^{13}C$ values were most likely driven by differences in the resource organic matter $\delta^{13}C$ values e.g. changes to MPB $\delta^{13}C$ values that were likely associated with lower mangrove C fixation/respiratory inputs following the mangrove mortality. Furthermore, such findings indicate that the reported substantial change to the mangrove benthic faunal assemblage following the mangrove loss (Harada et al., 2019) was probably driven more by modification of physical habitat structure than changes in the use of food resources". | Ln 392-397 |
| l.395: I don't understand "can reflect consumer tissues with little isotope effect" – how do the patterns in producers reflect patterns in consumers; shouldn't it be the other way round? | We have modified the line. "The $\delta^{13}C_{AA}$ patterns in producers, especially those of essential amino acids ($\delta^{13}C_{EAA}$), can be reflected in consumer tissues with little isotope effect". | Ln 386-387 |
| l.403: what is it that mostly affect MPB? Besides the biotic changes, we would expect much more light, and thus, higher evaporation and less water at the impacted than at the unimpacted site. This already will change MPB drastically. | Source of carbon and isotope fractionation can affect the isotope value of MPB. We have inserted "In our case, MPB $\delta^{13}C$ values changed significantly, likely due to changes to organic matter respiratory inputs and/or altered light environment and soil moisture contents that may change isotopic fractionation during carbon fixation". | Ln 346-348 |
| l.425: I do not understand how you derive these scenarios from the present study? I kind of agree with these potential scenarios (there might be other possibilities), but how does this relate to, how is this justified by, the present study? | These are likely scenarios and there might be other possibilities. What we have learned from this study is that biochemical changes can be reflected in the isotopic values of organisms. Multi-annual sampling can be used to track their changes overtime and such | Ln 424-425 |

| | isotopic information can be used to monitor biogeochemical changes in the future. It can be expected from this study that when the impacted forest is fully recovered, it would be isotopically similar to the unimpacted site. If the forest is unable to recover this may not be observed.

We have now inserted "While these are likely scenarios and there might be other possibilities, comparing the impacted forest and an adjacent unimpacted forest can help us quantify the recovery". | |
|---|---|---|
| Minor: l.410: omit "-" | We have now omitted "-" | |

[revised manuscript text omitted]

Traditional field methods such as measuring species composition to evaluate structure and functioning of ecosystems
75     can be time-consuming and expensive.   SIA of ecosystem components can   evalua functional aspects of element cycling and   food web in a cost-effective way ( Fry, 2006). To quantify food web dynamics, SIA of total organic matter (''bulk'') requires determination of the baseline isotope values of the food web, but this is difficult to achieve, particularly for complex
80    detrital food webs. Some of the uncertainties associated with bulk SIA have been clarified by compound-specific isotope analysis of amino acids (CSIA-AA). This technique has been increasingly used to assess pathways of energy transfer throughout food webs by distinguishing the effects of baseline isotope values and trophic transfer. Distinct isotopic fractionation between two groups of amino acids occurs with each trophic transfer. In general, non-essential and/or trophic AAs show large isotopic fractionation per trophic step, while essential and/or source AAs show little fractionation, reflecting
85    the baseline isotope values of the food web (Ishikawa et al., 2018; Larsen et al., 2013; Ohkouchi et al., 2017).

90

We investigated changes in C, N and S cycling associated with the Gulf of Carpentaria mangrove forest dieback (Duke et al., 2017), using a combination of traditional ecological survey techniques, bulk SIA, and CSIA-AA of  carbon. We hypothesised that the mortality of mangrove foundation species has changed the overall circulation of C, N and S elements and these biogeochemical changes would most likely be reflected in $\delta^{13}C$, $\delta^{15}N$ and $\delta^{34}S$ values of mangrove ecosystem
95    components such as mangrove plants, sediment and associated animals. We also tested the hypothesis that these isotopic compositions changed over time with the recovery of mangrove vegetation. $\delta^{13}C$, $\delta^{15}N$ and $\delta^{34}S$ values were measured for samples including mangroves, sediment and invertebrates collected in a comparative setting of impacted mangrove forest site and an adjoining unaffected reference forest site in the Gulf of Carpentaria, Australia.

**2 Material and Methods**

100 ### 2.1 Study site

The Gulf of Carpentaria in tropical Northern Australia is an extensive, shallow coastal gulf. The area mainly consists of low-lying wetlands and is largely inaccessible with little direct human activity. Mangroves are abundant in the area, but the dry climate limits the extent, diversity, and height of mangroves in the region (Asbridge et al., 2016). The wide tidal wetlands spread along the shoreline with high intertidal saltpans and saltmarsh covering more area than mangroves. Mangroves are
105 often distributed in the seaward margin, typically as a narrow strip and fronted by extensive, shallow mudflats. The distribution of mangroves in this region is associated with tidal and freshwater inundation, river discharge and regular sediment supply through freshwater input. Increased amounts of rainfall and associated flooding and sea level rise were responsible for recent mangrove extension in this region between 1987 and 2014 (Asbridge et al., 2016). *Avicennia marina* was the dominant mangrove species at the study site in Karumba, Gulf of Carpentaria (Fig 1A).

110

Over 7,000 ha of mangroves along ~ 1,000 km of the Gulf of Carpentaria coastline in Australia experienced mass mortality during the summer in 2015-16, (Duke et al., 2017), the most extensive mangrove forest dieback ever recorded due to natural causes (Sippo et al., 2018). At the same time, there were coinciding mangrove mass mortality events in Exmouth, Western Australia (Lovelock et al., 2017) and Kakadu National Park, Northern territory (Asbridge et al., 2019). The climate in the Gulf
115 region is wet-dry tropical with mean annual precipitation ranging from approximately 600 to 900 mm. Dry conditions prevail for six to eight months and most rainfall occurs between December and March (Bureau of Meteorology, see www.bom.gov.au).  The dieback  coincided with  a weak monsoon (low rainfall), combined with high vapor pressure deficit, and El Niño–Southern Oscillation-induced low -sea-levels (Duke et al., 2017; Lovelock et al., 2017; Harris et al., 2017). The
120 drought conditions  most likely  caused accumulative hydric, thermal and radiant stresses (Duke et al., 2017). In addition to low water availability, iron (Fe) toxicity due to a rapid mobilisation of sedimentary Fe and regional variability in groundwater flows may have also played a role in the dieback (Sippo et al., 2020a). The event led to the widespread death of mangrove trees in the region providing an unfortunate yet  unique opportunity to test tree mortality effects on biogeochemical and ecological functioning
125 of mangroves and capture recovery patterns.

Three field campaigns were conducted in August 2016 (8 months after the event), August 2017 (20 months after the event) and August 2018 (32 months after the event) in the winter dry-seasons in Karumba. Some local factors (e.g. river influence and localised groundwater flow paths) may have kept some mangroves from dying
130 back in the region. A forest that had suffered dieback (impacted) on the east of Norman  River outlet and an adjoining unaffected forest (unimpacted) on the west, provided the setting for comparisons.

 In order to assess differences between the two forests (impacted vs. unimpacted), as well as to capture trends from across the intertidal zone and to ensure that the physical-oceanographic conditions between the two forests were as similar as possible, three sampling transects (2 to 2.5 km apart) were set for each forest with the length of each transect being approximately 200 m (Fig. 1B, C). Each transect consists of six sampling plots (approx. 40m apart), namely forest edge (landward), high, mid, low, forest edge (seaward) and mudflat (Table 1). Samples from each plot were pooled for analysis. Due to logistical constraints and the presence of saltwater crocodiles, fieldwork was restricted to daytime, low tide and dry seasons.

**2.2 Samples**

 Since our focus was to measure recovery of mangrove vegetation and food-web, we monitored mangrove sapling/seedlings and stable isotopes of invertebrates during the period from 2016 to 2018. Mangrove and sediment samples were also collected but they are limited to 2018 (Table 1). Some of the SIA samples including invertebrates, mangrove and sediment collected in 2017 were used to measure the initial dieback reported in Harada et al. (2019).

During each field campaign, four common mangrove macroinvertebrate groups with different feeding modes were collected from each forest including a leaf-eating crab (*Parasesarma molluccensis* and/or *Episesarma* sp.), an algae-eating (deposit-feeding) crab (*Tubuca signata*), a grazer gastropod (*Telescopium telescopium*) and a filter-feeding bivalve (*Saccostrea sp,* an oyster). For each feeding group, 3 to 5 individuals at each of the sampling transects (n=3) within the forest were collected and muscle tissues were pooled for SIA.

In 2018,  Ffully developed green leaves of *A. marina* were collected from at about 1 to 1.5 m height, from 3 to 5 individual trees (1 to 3 leaves per tree) at each sampling plot, stored in plastic containers, then composited. In the impacted site,  leaves were collected from  regrowth of  trees that had survived. Leaf samples were washed thoroughly, rinsed with distilled water and the main vein was removed. Additionally, wood samples (n=2, 5 to 25cm diameter) were collected using a hand saw from stems at chest height from the mid intertidal zone of each forest. Dead trees were sampled at the impacted site. Two to three bulk SIA measurements were made from sapwood (2 to 3cm deep) of  each  sample and measurements were averaged. Wood samples were generally very low in S and not sufficient  for $\delta^{34}$S analysis.

In 2018, surface (<0.5 cm) sediment that represent the recent deposition and microphytobenthos (MPB) were  collected along  each transect . Additionally,  subsurface (0.5 to 20 cm) sediment samples (n=6) that represent long-term averages

165    were collected at the mid intertidal zone of each forest using a core sampler (5 cm in diameter and 20 cm deep. For $\delta^{13}C$

[revised manuscript text omitted]

An explanation for our observed $\delta^{34}$S pattern may be lower plant incorporation of sulfide-S in the impacted site and also in the higher intertidal zones where we expect that mangrove sediment is relatively more oxidised, and the production of sulfide may be lower due to lower sulfate reduction. ~~High wood $\delta^{34}$S values (16.6‰) and S content (0.31%) in the impacted forest may suggest degradation of wood by fungi and/or bacteria that incorporate seawater sulfate-S and increase overall wood $\delta^{34}$S values and S content. Such $\delta^{34}$S patterns have been reported in mangroves (Fry and Smith, 2002) and saltmarsh (Currin et al., 1995), where $\delta^{34}$S values of fresh organic matter evolved during degradation steps and gradually increased towards the $\delta^{34}$S value of seawater sulfate-S (i.e. 21‰).~~

**4.2 Sediment**

In healthy mangrove forests, the fate of C fixed by primary producers includes burial within the sediment, atmospheric emissions and outwelling to the ocean (Maher et al., 2018), but how mangrove mortality affects such processes is poorly understood. In most cases, C within in mangrove sediment decreases following forest loss due to degradation with increased $CO_2$ emissions (Otero et al., 2017; Adame et al., 2018). Lower TOC (%) and higher sediment $\delta^{13}$C values in the impacted forest  are probably related to sediment C loss and lower autochthonous  C inputs (i.e. leaf litter) following the mangrove mortality event. Consistent with this, the sediment N (%) and $\delta^{15}$N data showed a similar pattern suggesting N loss and degradation. The surface sediment  varied more than the subsurface fraction. One explanation for his is probably because  the surface  fraction is generally more aerobic  and  remineralization of organic matter occurs more rapidly (Burdidge, 2011). Sediment $\delta^{13}$C and $^{15}$N values can increase during degradation of sediment organic matter following mangrove loss (Adame and Fry, 2016Adame et al., 2018). Changes in sediment C and N may also be associated with root turnover. The MPB $\delta^{13}$C values significantly differed, with those from the impacted being higher  than the unimpacted . The higher values probably indicate lower respiratory inputs of $CO_2$ from mangroves (Maher et al., 2013b). Our findings here are consistent with the finding of Sippo et al. (2019) that changes to oceanic carbon

outwelling rates following mangrove loss are likely associated with a gradual loss of sediment carbon; similar to our finding

390 of increased sediment $\delta^{13}$C values in the impacted site, an isotope effect may have been due to loss of sediment mangrove C or replacement of mangrove peats with marine sediment.

**4.3 Fauna**

CNS isotopic compositions of consumers including an algae feeder, a grazer and a leaf feeder from the impacted site were  more enriched in $^{13}$C, $^{15}$N and $^{34}$S. These differences remained consistent  throughout the three

395 sampling of 2016, 2017 and 2018 . Consistent with the findings from mangrove leaves, MPB and soil, the data suggested substantial changes in cycling of CNS associated with the mangrove mortality event. At the impacted site, consumers were more $^{13}$C-enriched,  likely due to the loss of $^{13}$C-depleted mangrove organic matter. Consumer $\delta^{13}$C values can change due to changes to available organic matter, altered feeding dependencies as well as changes to organic matter $\delta^{13}$C values. In

400 our case,  MPB $\delta^{13}$C values  changed significantly, likely due to changes to organic matter respiratory inputs and/or altered light environment and soil moisture contents that may change isotopic fractionation during carbon fixation . The consumer $\delta^{13}$C values and their ranges at our study site are fairly consistent with the reported mangrove consumer $\delta^{13}$C values elsewhere (e.g. Lee, 2000; Bouillon et al., 2002; Demopoulos et al., 2007). The typical trophic enrichment factor for carbon isotope in small invertebrates is about +1‰ (Vander Zanden and Rasmussen,

405 2001; McCutchan et al., 2003). Lower consumer $\delta^{13}$C values  are generally associated with mangrove detritus that is depleted in $^{13}$C. typical mangrove leaf-eating sesarmid crab species  generally have tissue $\delta^{13}$C values within  about +5‰ from  mangrove detritus Bui and Lee, 2014). Higher $\delta^{13}$C values of consumers are generally tied to MPB.  Our MPB endmember $\delta^{13}$C values of -

410 25.2‰ for the unimpacted site and -21.5‰ for the impacted site did not match with the consumer $\delta^{13}$C values (around -15 to -14‰), suggesting our characterization of MPB endmember $\delta^{13}$C values was incomplete. This is probably because MPB can vary substantially within mangrove ecosystems (Bouillon et al., 2008) and consumers may be preferentially assimilating more $^{13}$C enriched fractions of MPB, for example, diatom and/or filamentous cyanobacteria that can range about -15 to -20‰ (Craig, 1953; Fry and Wainright, 1991). The leaf feeder were relatively depleted in $^{13}$C  with $\delta^{13}$C values of

415 about -21 to -18‰likely due to  some use of mangrove leaves.

Due to difficulties obtaining representative endmembers, mixing analysis  was not achieved

420 using this data to quantify feeding dependencies. However, MPB probably played an important dietary role in the

both forests because the difference in MPB $\delta^{13}C$ values between the two forests were reflected in the difference in the consumers $\delta^{13}C$ values between the two forests (Harada et al. 2019). Alternatively, the consumer data was used to help infer endmembers and assess feeding dependencies, e.g. Riekenberg et al. (2016). POM (-21.1‰) matched with the filter feeders and seemed to be an important food source for the all consumers in both forests. Mangrove leaves (-27 to -25‰) did not seem

425  to be an important source for the consumers with the lowest consumer being -22.9‰ at the unimpacted site and -20.0‰ at the impacted site. The consumers were generally higher than the POM with the highest consumer being -15.2‰ at the unimpacted site and -14.0‰ at the impacted site, suggesting that there was a substantial contribution from more $^{13}C$ enriched MPB.

430

Consistent with the mangrove leave $\delta^{34}S$ values, the cConsumer $\delta^{34}S$ values also indicated possible changes to S cycling. The consumer $\delta^{34}S$ values were generally higher in the impacted site (range 13.4 to 21.7‰) than in the unimpacted site (range 8.2 to 16‰) suggesting lower sulfate reduction with decreased sulfide inputs at the impacted site. Fixation of sulfate by phytoplankton occurs with a small isotope effect, around 1 to 2‰ (Fry, 2006), therefore , phytoplankton $\delta^{34}S$ values from the

435  coastal ocean are generally close to the seawater sulfate-S value of 21‰. so that $\delta^{34}S$ values of phytoplankton from the coastal ocean should be close to the seawater sulfate-S value of 21‰. MPB generally have lower $\delta^{34}S$ values than phytoplankton, e.g. with reported average values near 10‰ for MPB in a mangrove ecosystem (Harada et al., unpublished), likely due to some use of sedimentary sulfide-S (depleted in $^{34}S$). Our consumer mangrove leaf $\delta^{34}S$ values were lower than 21‰, suggesting some use of MPB as well as mangrove detritus. averaged 13.5‰ for the impacted site and 12.6‰ for the unimpacted site, lower

[revised manuscript text omitted]

705 Vander Zanden, M. J., and Rasmussen, J. B.: Variation in $\delta^{15}N$ and $\delta^{13}C$ trophic fractionation: Implications for aquatic food web studies, Limnology and Oceanography, 46, 2061-2066, 2001.

Wernberg, T., Bennett, S., Babcock, R. C., de Bettignies, T., Cure, K., Depczynski, M., Dufois, F., Fromont, J., Fulton, C. J., Hovey, R. K., Harvey, E. S., Holmes, T. H., Kendrick, G. A., Radford, B., Santana-Garcon, J., Saunders, B. J., Smale, D. A., Thomsen, M. S., Tuckett, C. A., Tuya, F., Vanderklift, M. A., and Wilson, S.: Climate-driven regime shift of a temperate
710 marine ecosystem, Science, 353, 169-172, 2016.

Werth, M., Mehltreter, K., Briones, O., and Kazda, M.: Stable carbon and nitrogen isotope compositions change with leaf age in two mangrove ferns, Flora-Morphology, Distribution, Functional Ecology of Plants, 210, 80-86, 2015.

715 **Table 1.** Spatial and temporal details of the sampling design (x = sampled).

| | Sampling plots along each transect | | | | | | Year | | |
|---|---|---|---|---|---|---|---|---|---|
| | Forest edge, landward | High | Mid | Low | Forest edge, seaward | Mudflat | 2016, 8 months after dieback | 2017, 20 months after dieback | 2018, 32 months after dieback |
| Mangrove seedling count | | | x | | | | x | x | x |
| Bulk SIA invertebrates* | | x | x | x | | | x | x | x |
| CSIA invertebrates* | | x | x | x | | | | x | |
| Bulk SIA mangrove leaves | x | x | x | x | x | | | | x |
| Bulk SIA mangrove wood | | | x | | | | | | x |
| Bulk SIA surface sediment | x | x | x | x | x | x | | | x |
| Bulk SIA subsurface sediment | | | x | | | | | | x |

*for invertebrate SIA, to gain sufficient sampling size, high, mid and low plots were pooled for analysis.

[revised manuscript text omitted]